# Alternating Diffusion for Proximal Sampling with Zeroth Order Queries

**Hirohane Takagi**[1,†,*]**, Atsushi Nitanda**[2,3,4,‡]

[1] Graduate School of Information Science and Technology, The University of Tokyo, Japan
[2] Institute of High Performance Computing, Agency for Science, Technology and Research, Singapore
[3] Centre for Frontier AI Research, Agency for Science, Technology and Research, Singapore
[4] College of Computing and Data Science, Nanyang Technological University, Singapore
[†] htakagi@is.s.u-tokyo.ac.jp
[‡] atsushi_nitanda@a-star.edu.sg

## Abstract

This work introduces a new approximate proximal sampler that operates solely with zeroth-order information of the potential function. Prior theoretical analyses have revealed that proximal sampling corresponds to alternating forward and backward iterations of the heat flow. The backward step was originally implemented by rejection sampling, whereas we directly simulate the dynamics. Unlike diffusion-based sampling methods that estimate scores via learned models or by invoking auxiliary samplers, our method treats the intermediate particle distribution as a Gaussian mixture, thereby yielding a Monte Carlo score estimator from directly samplable distributions. Theoretically, when the score estimation error is sufficiently controlled, our method inherits the exponential convergence of proximal sampling under isoperimetric conditions on the target distribution. In practice, the algorithm avoids rejection sampling, permits flexible step sizes, and runs with a deterministic runtime budget. Numerical experiments demonstrate that our approach converges rapidly to the target distribution, driven by interactions among multiple particles and by exploiting parallel computation.

## 1 Introduction

Sampling from probability distributions $\pi(x) \propto e^{-f(x)}$ is a fundamental task in statistics and machine learning, with applications in Bayesian posterior inference and score-based generative modeling. Methods like the Unadjusted Langevin Algorithm (ULA) (Roberts & Tweedie, 1996; Durmus & Moulines, 2017) and the Metropolis-Adjusted Langevin Algorithm (MALA) (Roberts & Rosenthal, 1998; Roberts & Stramer, 2002) are widely used. Their convergence under strong convexity assumptions has been established in sharp nonasymptotic terms (Dalalyan, 2017; Dwivedi et al., 2018), and it has further been shown that ULA enjoys exponential convergence in KL divergence under functional inequalities (Cheng & Bartlett, 2018; Vempala & Wibisono, 2019).

Beyond Langevin-type approaches, there has been growing interest in alternative sampling schemes with nonasymptotic guarantees. Among them, *proximal sampling* (Liang & Chen, 2023b) introduces an auxiliary distribution close to the target—typically the Gaussian convolution of $\pi$ (Lee et al., 2021)—and alternates conditional updates between the target and auxiliary variables. From a theoretical perspective, each update can be interpreted as alternating a forward heat-flow step and a reverse denoising step, which yields exponential convergence under suitable functional-inequality assumptions on the target distributions (Chen et al., 2022; Kook et al., 2024; Wibisono, 2025).

Despite this line of analysis, scalable implementations of proximal samplers remain challenging. Existing implementations rely on local optimization of $f$ combined with rejection sampling (Liang & Chen, 2023a;b; Fan et al., 2023; Liang & Chen, 2024), which necessitates small step sizes (i.e., weak convolution) to maintain acceptance and thus incurs many iterations and high overall cost.

---

*This work was conducted as part of a research internship at Agency for Science, Technology and Research (A⋆STAR), Singapore.

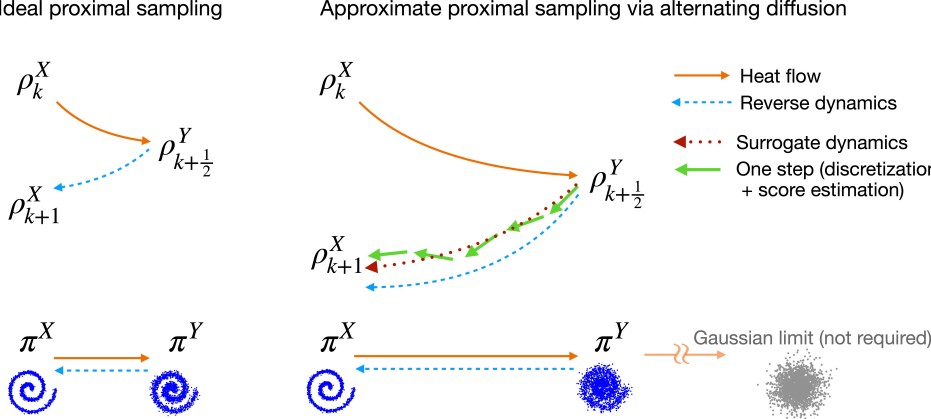

Figure 1: Illustration of the ideal proximal sampling (left) and our approximation (right). Heat flow and reverse dynamics are defined between $\pi^X$ and $\pi^Y$, but applied to intermediate $\rho$. Although these do not reach their targets in one step, the ideal version attains exponential convergence. Compared to the rejection sampling-based implementation of proximal samplers, our approach allows for larger step sizes (i.e., stronger convolution), which reduce the iterations to reach the target distribution.

These bottlenecks have spurred diffusion-based Monte Carlo, which simulates denoising stochastic differential equations (SDEs) (Huang et al., 2024a;b; He et al., 2024). This raises a natural question: *can proximal sampling be implemented in its theoretical form and in a scalable way, directly through Gaussian convolutions and diffusion processes, without relying on rejection sampling?*

Another practical consideration is the efficient sampling of many particles in parallel. In such settings, particle-based algorithms that introduce additional interaction terms or gradient-flow structures can promote faster mixing while maintaining diversity (Liu & Wang, 2016; Futami et al., 2020; Boffi & Vanden-Eijnden, 2023; Lu et al., 2024; Ilin et al., 2025). However, proximal sampling was originally formulated as a single-particle iterative framework, leaving a gap between its theoretical appeal and the practical demands of multi-particle sampling.

## 1.1 Contributions

In this work, we propose and analyze a diffusion-based approximation of proximal sampling. The key idea is to interpret the auxiliary samples as forming a Gaussian-mixture approximation and to exploit this structure for approximate time-dependent score estimation in the denoising dynamics. Our main contributions are as follows:

- We introduce a new algorithm that serves as an approximate proximal sampler. It is learning-free, gradient-free with respect to $f$, and rejection-free with a fixed runtime.

- We extend the theory of proximal sampling to show that our diffusion-based approximation inherits comparable convergence rates under suitable assumptions. This implies that our algorithm, which alternates perturbation and denoising, converges to the target distribution without requiring initialization from the Gaussian limit, unlike standard diffusion models.

- We provide empirical evidence that our method converges rapidly to representative targets compared to existing implementations of proximal sampler, and that its multi-particle extension improves both wall-clock efficiency and sample diversity.

## 2 Preliminaries

In this section, we provide a brief overview of proximal sampling, which underlies our proposed method, along with its convergence properties.

## 2.1 DEFINITIONS

Let $\mu$ and $\nu$ be two probability measures on $\mathbb{R}^d$ with $\mu \ll \nu$. We define the Kullback–Leibler (KL) divergence, the Rényi divergence of order $q \geq 1$, and the relative Fisher information as

$$H_\nu(\mu) = \int \log \frac{d\mu}{d\nu} \, d\mu, \quad \mathcal{R}_{q,\nu}(\mu) = \frac{1}{q-1} \log \int \left(\frac{d\mu}{d\nu}\right)^q d\nu, \quad J_\nu(\mu) = \int \left\|\nabla \log \frac{d\mu}{d\nu}\right\|^2 d\mu.$$

Note that setting $q = 1$ yields $\mathcal{R}_{1,\nu}(\mu) = H_\nu(\mu)$. We say that $\nu$ satisfies a log-Sobolev inequality (LSI) with constant $C_{\mathrm{LSI}} > 0$ if $H_\nu(\mu) \leq \frac{1}{2} C_{\mathrm{LSI}} J_\nu(\mu)$.

## 2.2 PROXIMAL SAMPLING

Let $f : \mathbb{R}^d \to \mathbb{R}$ be a potential function, and consider the target distribution $\pi^X(x) \propto \exp(-f(x))$. For a given step size $h > 0$, following Lee et al. (2021), we define a joint distribution on $\mathbb{R}^d \times \mathbb{R}^d$ as

$$\pi^{X,Y}(x, y) \propto \exp\left(-f(x) - \frac{\|x - y\|^2}{2h}\right). \tag{1}$$

Marginalizing over $y$ yields $\pi^X(x) \propto \int \pi^{X,Y}(x, y) \, dy$. On the other hand, marginalizing over $x$ defines a new distribution $\pi^Y$ as $\pi^Y(y) \propto \int \exp\left(-f(x) - \frac{1}{2h}\|x - y\|^2\right) dx$. This corresponds to the Gaussian convolution $\pi^Y = \pi^X * \mathcal{N}(0, hI_d)$.

The proximal sampler with step size $h$ iteratively updates a current particle $x_k \in \mathbb{R}^d$ as follows:

$$y_{k+\frac{1}{2}} \sim \mathcal{N}(x_k, hI_d) \tag{2}$$

$$x_{k+1} \sim \pi^{X|Y=y_{k+\frac{1}{2}}} \tag{3}$$

where $\pi^{X|Y=y_{k+1/2}}(x) \propto \exp(-f(x) - \frac{\|x - y_{k+1/2}\|^2}{2h})$. The restricted Gaussian oracle (RGO) (Lee et al., 2021) assumes access to an oracle which enables sampling (3) from $\pi^{X|Y=y_{k+1/2}}(x)$.

Several works have implemented the RGO (Liang & Chen, 2023a;b; Fan et al., 2023) through rejection sampling, achieving an expected complexity of $\tilde{O}(1)$ under regularity conditions on $f$. However, this requires choosing $h$ sufficiently small depending on $f$ and $d$, which in practice leads to many iterations, and the computational cost further fluctuates due to randomness.

## 2.3 CONNECTIONS TO DIFFUSION PROCESSES

Chen et al. (2022) established an improved convergence analysis of proximal sampling by interpreting each update through dynamics interpolated by an SDE, as illustrated in the left panel of Figure 1. The forward step (2) amounts to Gaussian convolution, which corresponds to evolving the heat equation or its associated SDE, for $t \in [0, h]$,

$$dZ_t = dB_t, \quad \partial_t \mu_t = \tfrac{1}{2} \Delta \mu_t, \tag{4}$$

where $B_t$ is a standard Brownian motion. Writing $P_t$ for the Gaussian convolution kernel $\mathcal{N}(0, tI_d)$, we have $\mu_t = \mu_0 P_t$, and in particular $\pi^Y = \pi^X P_h$. The backward step (3) can be viewed as the reverse operation of the forward step conditioned on $Z_h$. It is governed by the following SDE starting from $Z_0^\leftarrow = Z_h$ and the corresponding Fokker–Planck equation; for $t \in [0, h]$,

$$dZ_t^\leftarrow = \nabla \log(\pi^X P_{h-t})(Z_t^\leftarrow) \, dt + dB_t^\leftarrow, \quad \partial_t \mu_t^\leftarrow = -\mathrm{div}\left(\mu_t^\leftarrow \nabla \log(\pi^X P_{h-t})\right) + \tfrac{1}{2} \Delta \mu_t^\leftarrow, \tag{5}$$

where $B_t^\leftarrow$ denotes the backward Brownian motion. Letting $Q_t$ denote its transition kernel, we obtain $\mu_h^\leftarrow = \mu_0^\leftarrow Q_h = \int \pi^{X|Y=y}(x) \, \mu_0^\leftarrow(y) \, dy$. In particular, we have $\pi^X = \pi^Y Q_h$.

Hence the two steps in proximal sampling can be viewed as SDEs between $\pi^X$ and $\pi^Y$, with $P_t$ and $Q_t$ as their transition kernels. The proximal sampler applies the dynamics (4) and (5) with the start distributions $\rho_k^X := \mathrm{law}(x_k)$ and $\rho_{k+1/2}^Y := \mathrm{law}(y_{k+1/2})$ at the $k$-th iteration, respectively, which then evolve through Gaussian convolution or conditional integration. Although a single step does not reach $\pi^Y$ or $\pi^X$, iterating the forward–backward scheme leads to contraction towards the target distribution, as made precise in the following theorem.

**Theorem 1** (Chen et al. (2022), Theorem 3). *Assume that $\pi^X$ satisfies LSI with constant $C_{LSI}$. For any $h > 0$ and any initial distribution $\rho_0^X$, the $k$-th iterate $\rho_k^X$ of the proximal sampler with step size $h$ satisfies, for $q \geq 1$,*

$$\mathcal{R}_{q,\pi^X}(\rho_k^X) \leq \frac{\mathcal{R}_{q,\pi^X}(\rho_0^X)}{(1 + h/C_{LSI})^{2k/q}}. \tag{6}$$

Theorem 1 provides the exponential convergence guarantee of proximal sampling. Its proof follows by combining the forward and backward contraction properties at each step stated in the next lemma.

**Lemma 1** (Chen et al. (2022), Appendix A.4). *Assume that $\pi^X$ satisfies LSI with constant $C_{LSI}$ and $\rho \ll \pi^X$, $\rho' \ll \pi^Y$. For $q \geq 1$, we have*

$$\mathcal{R}_{q,\pi^Y}(\rho P_h) = \mathcal{R}_{q,\pi^X P_h}(\rho P_h) \leq \frac{\mathcal{R}_{q,\pi^X}(\rho)}{(1 + h/C_{LSI})^{1/q}}, \tag{7}$$

$$\mathcal{R}_{q,\pi^X}(\rho' Q_h) = \mathcal{R}_{q,\pi^Y Q_h}(\rho' Q_h) \leq \frac{\mathcal{R}_{q,\pi^Y}(\rho')}{(1 + h/C_{LSI})^{1/q}}. \tag{8}$$

In this paper, we aim to realize the proximal sampler by directly simulating the associated SDEs. In the forward direction, convolution with $\pi^X$ can be simulated exactly by injecting Gaussian noise, so the bound (7) is directly applicable. In contrast, the backward dynamics (5) cannot be simulated exactly, and our surrogate version of (5) will provide an approximation in place of (8).

## 3    APPROXIMATE MULTI-PARTICLE PROXIMAL SAMPLING

Building on the SDE interpretation in Section 2, we propose to approximate the backward dynamics by replacing $\pi^X$ with a surrogate distribution constructed from the current particles. This enables Monte Carlo estimation of the score function using only evaluations of $f$, without requiring gradient or rejection sampling. At a high level, each iteration of our diffusion-based proximal sampler consists of (i) evolving particles by the forward heat flow (4) and (ii) applying the surrogate dynamics, which time-discretizes and approximates the reverse flow (5).

### 3.1    SCORE ESTIMATION FOR SURROGATE DYNAMICS

We derive the update rule in the backward step, given the particles at the $k$-th iteration: $X_k = \{x_i\}_{i=1}^N$ and $Y_{k+1/2} = \{y_j\}_{j=1}^N$. We approximate the backward dynamics (5) by replacing $\pi^X$ with a surrogate distribution $\hat{q}_{k+1}(\cdot \mid Y_{k+1/2}, X_k)$ constructed from these particles, as follows:

$$\hat{q}_{k+1}(x \mid Y_{k+1/2}, X_k) \propto \frac{1}{N} \sum_{j=1}^N \pi^{X|Y=y_j}(x) \frac{\pi^Y(y_j)}{\hat{q}_{k+1/2}(y_j \mid X_k)}, \tag{9}$$

where $\hat{q}_{k+1/2}(y \mid X_k) = \frac{1}{N} \sum_i \mathcal{N}(y; x_i, hI_d)$. The conditional distribution with the reweighting term ensures that, as $N \to \infty$, sampling $y \sim \hat{q}_{k+1/2}$ recovers the target distribution $\pi^X$. By substituting the explicit form of $\pi^{X|Y=y_j}$ and $\pi^Y$ we obtain

$$\hat{q}_{k+1}(x \mid Y_{k+1/2}, X_k) \propto \frac{1}{N} \sum_{j=1}^N \frac{\exp\left(-f(x) - \frac{1}{2h}\|x - y_j\|^2\right)}{\hat{q}_{k+1/2}(y_j \mid X_k)}$$

$$\propto \frac{1}{N} \sum_{j=1}^N \frac{\mathcal{N}(x; y_j, hI_d)}{\hat{q}_{k+1/2}(y_j \mid X_k)} \exp(-f(x)) =: g_N^{k+1/2}(x) \exp(-f(x)), \tag{10}$$

where $g_N^{k+1/2}$ is the unnormalized density of an $N$-component weighted Gaussian mixture (see Appendix A.1 for the full derivation). $\hat{q}_{k+1}$ involves an inverse reweighting with respect to a Gaussian mixture $\hat{q}_{k+1/2}$. This reduces weights when $y_j$ is located in areas where $X_k$ is overly concentrated, while amplifying their importance in sparse regions. The resulting term, derived from the empirical particle system, may help prevent particle collapse and promote exploration. We later verify this effect in our experiments.

---

**Algorithm 1** Zeroth-Order Diffusive Proximal Sampler

---

**Input:** potential function $f : \mathbb{R}^d \to \mathbb{R}$, initial samples $\{x_0^{(i)}\}_{i=1}^N$ step size $h$, iterations $K$, diffusion steps $T$, number of interim samples $M$, noise schedule $\{\sigma_t^2\}_{t=0}^T$ with $\sigma_T^2 = h$ and $\sigma_0^2 = \sigma_{\min}^2$.

1:   $\triangleright$ All operations for $i = 1, \ldots, N$, $j = 1, \ldots, N$, and $l = 1, \ldots, M$ are evaluated in parallel.
2: **for** $k = 0, 1, \ldots, K-1$ **do**
      # Step 1: Forward heat flow (4)
3:      $y_{k+\frac{1}{2}}^{(j)} \leftarrow x_k^{(j)} + \sqrt{h}\,\xi_k^{(j)}$ with $\xi_k^{(j)} \stackrel{\text{i.i.d.}}{\sim} \mathcal{N}(0, I_d)$.
4:      Initialize $z_T^{(i)} \leftarrow x_k^{(i)} + \sqrt{h}\,\xi_k'^{(i)}$ with $\xi_k'^{(i)} \stackrel{\text{i.i.d.}}{\sim} \mathcal{N}(0, I_d)$.
      # Step 2: Surrogate version of reverse dynamics (5)
5:      **for** $t = T, T-1, \ldots, 1$ **do**
6:          Set $\Delta t \leftarrow \sigma_t^2 - \sigma_{t-1}^2$.
7:          Compute $\bar{\sigma}, m_{i,j} := m_j(z_t^{(i)}), w_{i,j} := w_j(z_t^{(i)})$ by (14) and set $w_{i,j} \leftarrow w_{i,j} / \sum_j w_{i,j}$.
8:          Draw $M$ samples $z_0^{(i,l)} \stackrel{\text{i.i.d.}}{\sim} \sum_j w_{i,j} \mathcal{N}(m_{i,j}, \bar{\sigma}^2)$.
9:          Compute $c_{i,l} := \exp\big(- f(z_0^{(i,l)})\big)$ and set $c_{i,l} \leftarrow c_{i,l} / \sum_l c_{i,l}$.
          # Update each particle using the Euler–Maruyama step
10:          $z_{t-1}^{(i)} \leftarrow z_t^{(i)} + \Delta t \sum_l c_{i,l}\big(z_0^{(i,l)} - z_t^{(i)}\big)/\sigma_t^2 + \sqrt{\Delta t}\,\xi_k''^{(i)}$ with $\xi_k''^{(i)} \stackrel{\text{i.i.d.}}{\sim} \mathcal{N}(0, I_d)$.
11:      **end for**
12:      Set $x_{k+1}^{(i)} \leftarrow z_0^{(i)}$.
13: **end for**
14: **return** $\{x_K^{(i)}\}_{i=1}^N$

---

The surrogate score is defined as $\hat{s}_t(Z_t^{\leftarrow}) := \nabla \log(\hat{q}_{k+1} P_{h-t})(Z_t^{\leftarrow})$, which serves as the drift term in the surrogate reverse dynamics, replacing $\nabla \log(\pi^X P_{h-t})(Z_t^{\leftarrow})$ in the backward dynamics (5). This expression can be rewritten in expectation form using Bayes' rule:

$$\hat{s}_t(z) = \mathbb{E}_{g_N^{k+1/2}(x_0|z)}\left[\frac{x_0 - z}{\sigma_t^2} e^{-f(x_0)}/C_t^k\right], \tag{11}$$

$$\text{where} \quad \sigma_t^2 = h - t, \quad C_t = \int g_N^{k+1/2}(x_0 \mid z) e^{-f(x_0)}\,\mathrm{d}x_0. \tag{12}$$

See Section A.1 for the derivation. Since $g_N^{k+1/2}(x_0)$ is a Gaussian mixture and the conditional law $z \mid x_0$ under the perturbation kernel is $\mathcal{N}(x_0, \sigma_t^2 I_d)$, Bayes' rule implies that the posterior distribution $g_N^{k+1/2}(x_0 \mid z)$ is itself a Gaussian mixture:

$$g_N^{k+1/2}(x_0|z) \propto \sum_{j=1}^N w_j(z)\,\mathcal{N}(x_0; m_j(z), \bar{\sigma}^2 I), \tag{13}$$

$$\text{where } \bar{\sigma}^2 = (h^{-1} + \sigma_t^{-2})^{-1},\ m_j(z) = \bar{\sigma}^2(h^{-1}y_j + \sigma_t^{-2}z),\ w_j(z) = \frac{\mathcal{N}(z; y_j, (h + \sigma_t^2)I_d)}{\hat{q}_{k+1/2}(y_j \mid X_k)}. \tag{14}$$

The detailed derivation can be found in Section A.2. This formulation enables practical score estimation via Monte Carlo sampling from the Gaussian mixture without requiring model training.

## 3.2 ALGORITHM AND COMPLEXITY

The surrogate score (11) involves an expectation with respect to the Gaussian-mixture posterior (13). In practice, we approximate this expectation using Monte Carlo estimation by drawing $M$ samples for each particle. Combining the forward step (Gaussian perturbation) and the discretized surrogate reverse dynamics with $T$ denoising steps, we obtain the iterative sampling procedure that approximates the proximal sampling. It is worth emphasizing that the proposed method relies only on a zeroth-order oracle of $f$ and does not require gradient information, unlike gradient-based sampling methods such as Langevin Monte Carlo. A complete description of the algorithm is provided in Algorithm 1.

Regarding computational cost, a straightforward implementation would require $KTMN$ evaluations of the potential $f$ for $K$ outer iterations. This complexity can be further reduced to $KT$ under the assumption of a parallel oracle that can evaluate $f$ simultaneously on multiple samples. Such a design naturally exploits parallel computation, making the method efficient in modern computing environments.

## 4 THEORETICAL ANALYSIS

In this section, we provide a theoretical analysis that essentially quantifies the deviation from the ideal proximal sampling caused by time and space discretization. Our method splits the macro step size $h$ into $T$ substeps of length $\eta = h/T$ for the backward dynamics. For each $\ell \in \{1, \ldots, T\}$, let $t_\ell := (\ell - 1)\eta$ be the substep start time, and let $\mu_{t_\ell}$ denote the current law at time $t_\ell$. The following lemma establishes the contraction of the KL divergence in a single outer iteration of Algorithm 1, up to an error due to time discretization. We assume that $\pi^X$ satisfies an LSI with constant $C_{\text{LSI}}$. Focusing on iteration $k$, we write $\rho_k^X := \text{law}(x_k^{(i)})$ and $\rho_{k+1/2}^Y := \text{law}(y_{k+1/2}^{(j)})$. A complete description of the assumptions and proofs is deferred to Appendix A.

**Lemma 2** (Discretization-only one-step bound). *Assume $N, M \to \infty$, so the backward update uses the exact score. Suppose further that within each substep $\ell$, the temporal variation of the score between $t_\ell$ and $t_\ell + t$ ($t \in [0, \eta]$) integrating over $\mu_{t_\ell + t}$ is bounded by $L_{\ell,t}^2 t^2$, the score of the reference measure $\nu_{t_\ell}$ is Lipschitz in space with constant $L_{\nu,\ell}$, and there exists a uniform entropy bound $\bar{H}^{(k)} > 0$ along the substep path. Then for $u \geq 1$,*

$$H_{\pi^X}(\rho_{k+1}^X) \leq \frac{H_{\pi^Y}(\rho_{k+1/2}^Y)}{(1 + h/C_{LSI})^{1-1/(2u^2)}} + \alpha_u \Lambda_1^{(k)},$$

$$\text{where} \quad \alpha_u := 2u^4 C_{LSI}\left((1 + h/C_{LSI})^{1/(2u^2)} - 1\right),$$

$$\Lambda_1^{(k)} := 4\eta^2 L_{\nu,*}^4 (C_{LSI} + h)\bar{H}^{(k)} + \eta\, d\, C^{(k)},$$

$$C^{(k)} := 2\eta L_{\nu,*}^3 + L_{\nu,*}^2 + \eta L_s^2/d, \text{ with } L_{\nu,*} := \sup_\ell L_{\nu,\ell} \text{ and } L_s := \sup_{\ell,t} L_{\ell,t}.$$

The additional discretization error term in Lemma 2 scales as $O(h/T)$ with respect to the number of substeps $T$. Therefore, if $T$ is chosen appropriately, the backward step essentially inherits the convergence rate of the ideal proximal sampler in (8).

When $N, M < \infty$, we assume that at each substep start $t_\ell$ the score estimator admits the bound $\Lambda_2^{(k)} := 2\sup_{\ell=1,\ldots,T} \mathbb{E}_{\mu_{t_\ell}}[\|\hat{s}_{N,M}^{(k)}(\cdot, t_\ell) - s_{t_\ell}^{(k)}(\cdot)\|^2]$, where $s_{t_\ell}$ is the true score at $t_\ell$ and $\hat{s}_{N,M}^{(k)}(\cdot, t_\ell)$ is the Monte Carlo estimator. The following result incorporates this condition.

**Proposition 1** (Main one-step bound with split errors). *Let $u \geq 1$. Under the assumptions of Lemma 2 with finite $N, M$, and the score estimation bound, the $k$-th iterate of Algorithm 1 satisfies*

$$H_{\pi^X}(\rho_{k+1}^X) \leq \frac{H_{\pi^Y}(\rho_{k+1/2}^Y)}{(1 + h/C_{LSI})^{1-1/(2u^2)}} + \alpha_u(\Lambda_1^{(k)} + \Lambda_2^{(k)}).$$

Proposition 1, combined with the KL contraction in the forward step (Lemma 1, (7)), shows that $H_{\pi^X}(\rho_k^X)$ decreases geometrically in $k$, up to the accumulated approximation errors. Indeed, combining the forward contraction with the approximate backward bound yields

$$H_{\pi^X}(\rho_{k+1}^X) \leq \frac{1}{r} H_{\pi^X}(\rho_k^X) + \alpha_u(\Lambda_1^{(k)} + \Lambda_2^{(k)}),$$

where $r := (1 + h/C_{LSI})^{2-1/(2u^2)} > 1$. Unrolling this inequality gives, for any $k \geq 1$,

$$H_{\pi^X}(\rho_k^X) \leq \frac{1}{r^k} H_{\pi^X}(\rho_0^X) + \alpha_u \sum_{l=0}^{k-1} \frac{\Lambda_1^{(l)} + \Lambda_2^{(l)}}{r^{k-1-l}}.$$

In particular, if the per-step error satisfies the uniform bound $\Lambda_1^{(j)} + \Lambda_2^{(j)} \le \Lambda$ for all $j \ge 0$, then using $\sum_{l=0}^{k-1} r^{-l} \le \frac{r}{r-1}$ yields

$$H_{\pi^X}(\rho_k^X) \;\le\; \frac{1}{r^k}\, H_{\pi^X}(\rho_0^X) \;+\; \frac{\alpha_u r}{r-1}\, \Lambda.$$

Consequently, to ensure $H_{\pi^X}(\rho_k^X) \le \varepsilon$, it suffices to choose

$$k \;\ge\; \frac{\log\big(2 H_{\pi^X}(\rho_0^X)/\varepsilon\big)}{\log r} = \frac{\log\big(2 H_{\pi^X}(\rho_0^X)/\varepsilon\big)}{2(1 - 1/(2u^2))\log(1 + h/C_{LSI})} \quad \text{and} \quad \Lambda \;\le\; \frac{\varepsilon(r-1)}{2\alpha_u r}.$$

Equivalently, this yields the iteration complexity $k = O\big(\log(H_{\pi^X}(\rho_0^X)/\varepsilon)/\log r\big)$ under the admissible error level $\Lambda = O\big(\varepsilon(r-1)/(\alpha_u r)\big)$.

As for the score estimation error $\Lambda_2^{(k)}$, it reflects the Monte Carlo error arising from the finite sample sizes $N$ and $M$. As shown in Section A.4, for instance, under bounded conditional variances of $\pi^X$ we obtain $O(1/N)$ error, and under bounded expectations of $e^{4f}$ and finite fourth moments under the Gaussian mixture proposals we obtain $O(1/M)$ error, yielding $\Lambda_2^{(k)} = O(1/N + 1/M)$.

From Theorem 1, the ideal proximal sampler requires about $(2\log(1 + h/C_{LSI}))^{-1}$ outer iterations times a logarithmic factor in the initial divergence. When $C_{LSI} \gg h$, this factor is approximated by $\tilde{O}(C_{LSI}/h)$, suggesting that larger $h$ is favorable. While rejection-sampling implementations of RGO suffer from an upper bound on $h$, our method can take $h$ large as long as $T = O(h)$ to control the discretization error. This does not change the overall computational cost (outer iterations $\times$ $T$ steps), but allows $h$ to reflect the global structure of $\pi^X$ such as inter-mode distances rather than only the local smoothness of $f$, which we confirm to be practically advantageous in experiments.

## 5 EXPERIMENTS

We design two experiments to showcase the strengths of our algorithm as an approximation to proximal sampling. First, we revisit the Gaussian Lasso mixture (Liang & Chen, 2023b) to test whether our method accelerates convergence beyond RGO-based proximal sampling, especially via parallel particle updates. Second, we study uniform distributions over bounded, nonconvex, and disjoint domains, comparing our method with In-and-Out (Kook et al., 2024), the proximal sampler for uniform distributions originally analyzed for convex bodies. These experiments respectively demonstrate the benefits of larger step sizes and the applicability of our approach when gradients are unavailable.

### 5.1 GAUSSIAN LASSO MIXTURE

**Setup.** Following the experimental setting in Liang & Chen (2023b), we set the target distribution as a Gaussian Lasso mixture:

$$\pi^X(x) = \frac{\sqrt{\det Q}}{2\sqrt{(2\pi)^d}} \exp\left(-\frac{1}{2}(x-1)^\top Q(x-1)\right) + 2^{d-1} \exp\left(-\|4x\|_1\right)$$

where $Q = USU^\top$, $d = 5$, $S = \mathrm{diag}(14, 15, 16, 17, 18)$, and $U$ is an arbitrary orthogonal matrix.

We compare against the RGO baseline with 100 chains and step size $h=1/135$, exactly matching the experimental setting of Liang & Chen (2023b), where it outperformed ULA and MALA. Our method uses step size $h=1/10$ under two settings: $N=100$ interacting particles, or $N=1$ run with 100 chains. For comparability, we thin the RGO baseline by grouping every 10 single-step updates into one iteration, so that the parallel $f$-evaluation cost is comparable to ours.

Convergence is measured by KL divergence to the target distribution, estimated using Büth et al. (2025) with a fixed reference of 1000 particles from a long RGO run. At each evaluation, 1000 particles are pooled across 10 successive iterations. The histogram in the right panel of Figure 3 shows this reference distribution. All experiments are repeated with 10 random seeds, reporting the mean and variance of the KL estimate. Detailed settings are given in Table 1 and Section B.

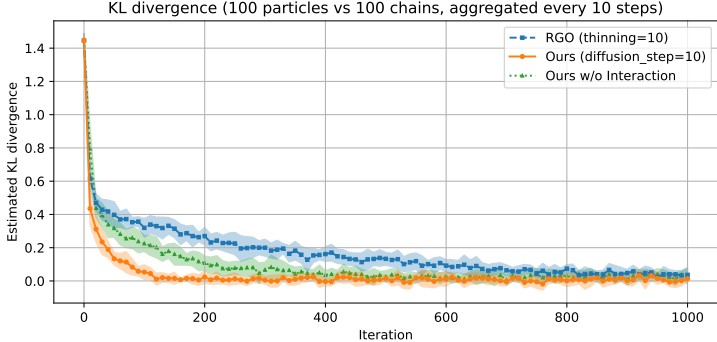

Figure 2: Convergence of estimated KL divergence, averaged over 10 random seeds with shaded areas indicating variances. Our method (orange) outperforms both the proximal sampler with RGO (blue) and an ablated variant of our algorithm without particle interactions (green). It achieves the same accuracy as RGO in about $10\times$ fewer iterations ($100\times$ faster when accounting for thinning).

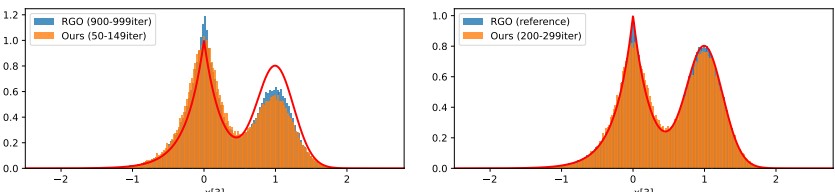

Figure 3: One-dimensional marginals of $\pi^X$ along the third coordinate. The red curve is the ground-truth. Our method around 100 iterations (left, orange) already matches the RGO-based sampler at $\sim 1000$ iterations (left, blue), and with only 200–300 iterations (right, orange) it closely aligns with the reference obtained from a sufficiently long run following Liang & Chen (2023b) (right, blue).

**Results.**    Figure 2 shows the convergence curves. Our method converges substantially faster than the RGO baseline. With a step size $13.5$ times larger, our method reaches comparable KL divergence in $\sim 100$ iterations, while the proximal sampler needs $\sim 950$ iterations ($\approx 9500$ RGO updates). Our method also surpasses the ablated variant with $N{=}1$ independent parallel chains, indicating that particle interactions are essential to accelerate mixing. The marginals in Figure 3 indicate that our method is already approaching the target by $\sim 100$ iterations (vs. $\sim 950$ for RGO), and it closely matches the long-run reference by $\sim 250$ iterations. A more detailed observation is deferred to Section B (Figure 5), which shows that increasing the step size $h$ leads to faster convergence.

**Discussion.**    These results demonstrate that relaxing the stringent step size restriction of proximal sampling yields a practical gain of nearly an order of magnitude in convergence speed. The benefit is amplified when leveraging multiple interacting particles, which facilitate more efficient exploration of the mixture components. This effect can be explained by the surrogate target distribution in (9), which re-weights via $(\hat{q}_{k+1/2}(y \mid X_k))^{-1}$ to down-weight overpopulated regions and encourage exploration of sparser ones, thereby accelerating convergence beyond the step-size effect alone.

## 5.2 UNIFORM DISTRIBUTIONS ON BOUNDED DOMAINS

**Setup.**    We evaluate our method on uniform sampling over a bounded, nonconvex, and disjoint domain $K \subset \mathbb{R}^3$, and compare it with the In-and-Out (Kook et al., 2024). In-and-Out proposes $y_{i+1} \sim \mathcal{N}(x_i, hI_d)$ and repeatedly resamples $x_{i+1} \sim \mathcal{N}(y_{i+1}, hI_d)$ until $x_{i+1} \in K$ or a retry threshold $R$ is reached. It converges under warm starts, but practical efficiency requires a convex $K$.

We take $K$ as the union of two disjoint solid tori in $\mathbb{R}^3$. $T_1$ is a torus centered at $(10, 0, 0)$ with major radius 10 and minor radius 1, and $T_2$ is a torus centered at $(-13, 0, 0)$ with major radius 3 and minor radius 1. We denote $K = T_1 \cup T_2$. Particles start from $\mathcal{N}(0, I_3)$. Our sampler uses

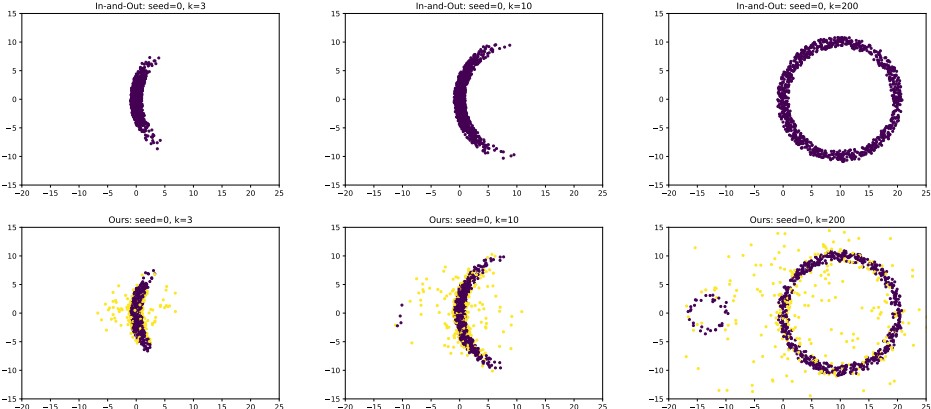

Figure 4: Empirical distributions at $k = 3, 10, 200$ iterations with seed $= 0$ on the two-tori domain. In-and-Out (top row) finds $T_1$, which overlaps with the initial standard Gaussian distribution, but fails to reach $T_2$. Our method (bottom row) generates some particles outside the domain but gradually drives particles toward $T_2$, demonstrating its ability to explore both components.

the potential $f(x) = 0$ if $x \in K$ and $100$ otherwise, inducing an approximate uniform law. We run $1,000$ particles and monitor their occupancies, visualized from the third coordinate direction. Details of the experimental setting are given in Table 2.

**Results.** Figure 4 shows that In-and-Out converges to uniformity inside the near torus $T_1$, consistent with its guarantee under warm starts, but fails to reach the distant torus $T_2$. Our method first fills the near torus, then transitions to the remote one, where many particles eventually accumulate.

**Discussion.** This experiment confirms that our algorithm works without gradients and can explore disconnected modes where exact proximal steps stagnate. The noisy score approximation facilitates such transitions, akin to effects observed in diffusion-based black-box optimization (Lyu et al., 2025). Other constrained-domain samplers include Projected Langevin Monte Carlo (Bubeck et al., 2018), MYULA (Brosse et al., 2017), and penalized Langevin dynamics based on distance to the constraint set (Gurbuzbalaban et al., 2024), as well as proximal samplers for convex bodies using projection or separation oracles (Dang & Liang, 2025). All these methods critically rely on projection- or separation-type operations and are therefore limited to simple convex bodies where such mappings are computable. By contrast, our sampler only requires a membership oracle or a simple outside penalty, making the zeroth-order oracle framework applicable to a broader range of domains.

## 6 CONNECTIONS TO OTHER SAMPLING METHODS

Diffusion-based Monte Carlo methods transport samples from a Gaussian initialization to the target distribution. Some rely on training models for score estimation (Vargas et al., 2023; Richter & Berner, 2024), while others drive auxiliary samplers (Huang et al., 2024a; He et al., 2024). Several works also develop acceleration and correction techniques within this paradigm (Lu et al., 2022; Kim & Ye, 2023; Li et al., 2024). Beyond diffusion-based pushforwards, related approaches construct explicit density paths (Fan et al., 2024; Guo et al., 2025).

Our approach also specifies and employs SDE dynamics between two fixed distributions, but unlike the one-way pushforward paradigm, it repeatedly applies these dynamics. This removes the restriction of restarting from a Gaussian base by using a variance-expanding (VE) diffusion. In the context of variance-preserving (VP) diffusions, sampling error partly arises from the discrepancy between the Gaussian equilibrium and the distribution obtained after finite-time mixing (Li et al., 2023; Pierret & Galerne, 2025). Our analysis suggests that, even though the VE process does not reach the Gaussian equilibrium in any bounded horizon, convergence can still be ensured by alternating finite-time noise addition and denoising dynamics through repeated cycles.

A particularly relevant comparison is with ZOD-MC (He et al., 2024), a diffusion-based method that performs denoising from a Gaussian initialization. In ZOD-MC, the outer loop transports samples while an inner proximal sampler is used for score estimation. Our method inverts this structure: instead of nesting a sampler inside a pushforward loop, we simulate proximal-style SDE dynamics directly. Moreover, unlike ZOD-MC, which requires access to the minimizer of the potential—or, in practice, gradient information to locate local solutions for proximal updates—our algorithm operates solely with zeroth-order oracle information. Another related line is SLIPS (Grenioux et al., 2024), which alternates between denoising noisy observations and updating auxiliary variables. This alternating scheme resembles ours, but the overall iteration follows one-way dynamics that converge to the target distribution as the time horizon grows.

Finally, our method is also close in spirit to Diffusive Gibbs Sampling (DiGS) (Chen et al., 2024), which performs Gibbs updates by alternating perturbation and denoising, updating both the state and an auxiliary variable at each step. However, the designs differ: DiGS employs VP diffusion with auxiliary samplers such as MALA and provides no convergence guarantees, whereas our method uses VE diffusion consistent with proximal sampling, yields provable contraction guarantees, and exploits parallel multi-particle computation for lightweight score estimation.

## 7 CONCLUSION

We have introduced a diffusion-based approximation of proximal sampling, which simulates the backward SDE using only zeroth-order information of $f$. The key idea is to approximate the intermediate particle distribution by a Gaussian mixture, enabling score estimation without auxiliary samplers or additional model training.

Our method remains within the proximal sampling framework, while also being closely connected to diffusion-based Monte Carlo methods. Our theoretical analysis shows that it achieves comparable convergence rates when discretization and score estimation errors are properly controlled. Unlike rejection-sampling implementations of RGO, which are constrained to small step sizes by local properties of $f$, our algorithm may accommodate larger step sizes that reflect global features such as inter-mode distances.

Finally, we incorporate interaction terms among multiple particles, which empirically accelerate convergence. A complete theoretical characterization of these interactions, together with efficient parameter tuning across iterations, remains an important direction for future work. More generally, understanding how such empirical gains scale to higher-dimensional settings remains an open challenge, as the curse of dimensionality is a well-known issue in the broader context of zeroth-order methods and rejection sampling.

ACKNOWLEDGMENTS

This research is supported by the National Research Foundation, Singapore and the Ministry of Digital Development and Information under the AI Visiting Professorship Programme (award number AIVP-2024-004). Any opinions, findings and conclusions or recommendations expressed in this material are those of the author(s) and do not reflect the views of National Research Foundation, Singapore and the Ministry of Digital Development and Information.

## REPRODUCIBILITY STATEMENT

We provide complete assumptions and full proofs of all theoretical results in Appendix A. The detailed settings of our experimental evaluations are described in Section 5 and Section B. In addition, the source code for reproducing the empirical results is available at `https://github.com/htkg/zod-ps`.

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

# A    PROOFS AND TECHNICAL DETAILS

## A.1    DERIVATION OF THE SURROGATE SCORE ESTIMATOR

We begin by substituting $\pi^X$ with

$$\hat{q}_{k+1}(x \mid Y_{k+\frac{1}{2}}, X_k) \propto \frac{1}{N}\sum_{j=1}^{N} \pi^{X|Y=y_j}(x)\,\frac{\pi^Y(y_j)}{\hat{q}_{k+\frac{1}{2}}(y_j \mid X_k)},$$

where $\hat{q}_{k+1/2}(y \mid X_k) = \frac{1}{N}\sum_i \mathcal{N}(y; x_i, hI_d)$. This follows from the fact that the forward heat flow (4) transports the empirical Dirac mixture $\frac{1}{N}\sum_i \delta_{x_i}$ to the Gaussian mixture $\hat{q}_{k+1/2}$.

Using

$$\pi^{X|Y=y_j}(x) = \frac{\exp\!\big(-f(x) - \frac{1}{2h}\|x - y_j\|^2\big)}{\int \exp\!\big(-f(u) - \frac{1}{2h}\|u - y_j\|^2\big)\,\mathrm{d}u},$$

$$\pi^Y(y_j) \propto \int \exp\!\big(-f(u) - \frac{1}{2h}\|u - y_j\|^2\big)\,\mathrm{d}u,$$

we obtain

$$\hat{q}_{k+1}(x \mid Y_{k+\frac{1}{2}}, X_k) \propto \frac{1}{N}\sum_{j=1}^{N} \frac{\exp\!\big(-f(x) - \frac{1}{2h}\|x - y_j\|^2\big)}{\hat{q}_{k+\frac{1}{2}}(y_j \mid X_k)}$$

$$\propto \left(\frac{1}{N}\sum_{j=1}^{N} \frac{\mathcal{N}(x; y_j, hI_d)}{\hat{q}_{k+1/2}(y_j \mid X_k)}\right)\exp(-f(x)) =: g_N^{k+1/2}(x)\exp(-f(x)),$$

where $g_N^{k+1/2}$ is the unnormalized density of an $N$-component weighted Gaussian mixture. As $N \to \infty$,

$$\lim_{N\to\infty} g_N^{k+1/2}(x) = \int \frac{\mathcal{N}(x; y, hI_d)}{\hat{q}_{k+1/2}(y \mid X_k)}\hat{q}_{k+1/2}(y \mid X_k)\mathrm{d}y = \mathrm{const},$$

and therefore $\hat{q}_{k+1}(\cdot \mid Y_{k+1/2}, X_k) \to \pi^X(\cdot)$.

Since $\hat{q}_{k+1}$ can be regarded as the product of (i) a weighted Gaussian mixture term and (ii) the exponential factor $\exp(-f)$, the surrogate score function $\hat{s}_t(Z_t^{\leftarrow}) := \nabla \log(\hat{q}_{k+1}P_{h-t})(Z_t^{\leftarrow})$ can be computed using only the particles and evaluations of $f$. Following the derivation in Lyu et al. (2025), we obtain

$$\hat{s}_t(z) = \nabla \log\left(\int g_N^{k+1/2}(x_0)\,p(z \mid x_0)\,e^{-f(x_0)}\mathrm{d}x_0\right)$$

$$= \frac{\int g_N^{k+1/2}(x_0)\,\nabla p(z \mid x_0)\,e^{-f(x_0)}\mathrm{d}x_0}{\int g_N^{k+1/2}(x_0)\,p(z \mid x_0)\,e^{-f(x_0)}\mathrm{d}x_0}.$$

Here

$$p(z \mid x_0) = \frac{1}{(2\pi\sigma_t^2)^{d/2}}\exp\left(-\frac{\|z - x_0\|^2}{2\sigma_t^2}\right), \quad \sigma_t^2 = h - t,$$

so that

$$\nabla p(z \mid x_0) = p(z \mid x_0)\,\frac{x_0 - z}{\sigma_t^2}.$$

Substituting into the above expression yields

$$\hat{s}_t(z) = \frac{\int g_N^{k+1/2}(x_0)\,p(z \mid x_0)\,\frac{x_0 - z}{\sigma_t^2}e^{-f(x_0)}\mathrm{d}x_0}{\int g_N^{k+1/2}(x_0)\,p(z \mid x_0)\,e^{-f(x_0)}\mathrm{d}x_0}$$

$$= \frac{\int g_N^{k+1/2}(x_0 \mid z)\,\frac{x_0 - z}{\sigma_t^2}e^{-f(x_0)}\mathrm{d}x_0}{\int g_N^{k+1/2}(x_0 \mid z)\,e^{-f(x_0)}\mathrm{d}x_0},$$

which is exactly the score function of the surrogate dynamics (11).

*Remark* 1. Lyu et al. (2025) consider distributional black–box optimization for maximizing $f$. When $p^k(x_0)$ is a Gaussian mixture, they show that setting $p^{k+1}(x_0) \propto p^k(x_0) \exp(-f(x_0)/\lambda)$ as the denoised distribution at time 0 in VP diffusion, the score at each time $t$ can be computed directly from samples of the posterior Gaussian mixture, without any auxiliary sampler.

A structurally similar mechanism appears in our particle-based proximal sampling. To sample from $\pi^X(x) \propto \exp(-f(x))$ (in practice, by moving samples from $\hat{q}_{k+1/2}(y \mid X_k)$ toward $\hat{q}_{k+1}(x \mid Y_{k+1/2}, X_k)$), we place the denoised distribution $g_N^{k+1/2}(x) \exp(-f(x))$, where $g_N^{k+1/2}$ is the Gaussian mixture obtained by smoothing the empirical particles $Y_{k+1/2}$. The score identity yields a backward update that uses only zeroth-order evaluations of $f$.

## A.2 DERIVATION OF THE (UNNORMALIZED) GAUSSIAN MIXTURE POSTERIOR

When sampling from an $N$-component Gaussian mixture, we first draw a component index according to the relative weights and then sample from the corresponding Gaussian component. For this reason, we often ignore constant multiplicative factors of the mixture distribution for simplicity.

We write $g_N^{k+1/2}(x_0) \propto \sum_j \alpha_j \mathcal{N}(x_0; y_j, hI_d)$ with $\alpha_j := 1/\hat{q}_{k+1/2}(y_j \mid X_k)$. By Bayes' rule,

$$
g_N^{k+\frac{1}{2}}(x_0 \mid z) = \frac{g_N^{k+\frac{1}{2}}(x_0)\, p(z \mid x_0)}{\int g_N^{k+\frac{1}{2}}(u)\, p(z \mid u)\, \mathrm{d}u}
$$

$$
= \frac{\sum_{j=1}^N \alpha_j \mathcal{N}(x_0; y_j, hI_d)\, \mathcal{N}(z; x_0, \sigma_t^2 I_d)}{\sum_{j=1}^N \alpha_j \int \mathcal{N}(u; y_j, hI_d)\, \mathcal{N}(z; u, \sigma_t^2 I_d)\, \mathrm{d}u}
$$

$$
\propto \sum_{j=1}^N \alpha_j \mathcal{N}(x_0; y_j, hI_d)\, \mathcal{N}(z; x_0, \sigma_t^2 I_d). \tag{15}
$$

We now complete the square in the exponent of the product $\mathcal{N}(x_0; y_j, hI_d)\, \mathcal{N}(z; x_0, \sigma_t^2 I_d)$ viewed as a function of $x_0$. Using the identity

$$
\frac{1}{2h}\|x_0 - y_j\|^2 + \frac{1}{2\sigma_t^2}\|z - x_0\|^2 = \frac{1}{2\bar{\sigma}^2}\|x_0 - m_j(z)\|^2 + \frac{1}{2(h + \sigma_t^2)}\|z - y_j\|^2,
$$

with

$$
\bar{\sigma}^2 := \left(h^{-1} + \sigma_t^{-2}\right)^{-1} = \frac{h\,\sigma_t^2}{h + \sigma_t^2}, \quad m_j(z) := \bar{\sigma}^2\left(h^{-1}y_j + \sigma_t^{-2}z\right) = \frac{\sigma_t^2}{h + \sigma_t^2}y_j + \frac{h}{h + \sigma_t^2}z,
$$

we obtain the product-of-Gaussians factorization

$$
\mathcal{N}(x_0; y_j, hI_d)\, \mathcal{N}(z; x_0, \sigma_t^2 I_d) = \mathcal{N}(z; y_j, (h + \sigma_t^2)I_d)\, \mathcal{N}(x_0; m_j(z), \bar{\sigma}^2 I_d).
$$

Substituting this into (15) yields

$$
g_N^{k+\frac{1}{2}}(x_0 \mid z) \propto \sum_{j=1}^N \alpha_j \mathcal{N}(z; y_j, (h + \sigma_t^2)I_d)\, \mathcal{N}(x_0; m_j(z), \bar{\sigma}^2 I_d).
$$

Therefore $g_N^{k+1/2}(x_0 \mid z)$ is again a Gaussian mixture with a common covariance $\bar{\sigma}^2 I_d$ and updated means $m_j(z)$. Writing the relative weights as

$$
w_j(z) := \alpha_j \mathcal{N}(z; y_j, (h + \sigma_t^2)I_d) = \frac{1}{\hat{q}_{k+1/2}(y_j \mid X_k)} \mathcal{N}(z; y_j, (h + \sigma_t^2)I_d),
$$

we obtain the desired posterior decomposition

$$
g_N^{k+\frac{1}{2}}(x_0 \mid z) \propto \sum_{j=1}^N w_j(z)\, \mathcal{N}(x_0; m_j(z), \bar{\sigma}^2 I_d).
$$

Finally, note that using $W(z) = \sum_j w_j(z)$, the normalized posterior distribution can be written as

$$
g_N^{k+\frac{1}{2}}(x_0 \mid z) = \frac{1}{W(z)} \sum_{j=1}^N w_j(z)\, \mathcal{N}(x_0; m_j(z), \bar{\sigma}^2 I_d).
$$

## A.3 ANALYSIS FOR TIME-DISCRETIZATION ERROR

Our algorithm simulates the surrogate version of diffusion process (5) by dividing the dynamics over horizon $h$ into $T$ steps. We first analyze the discretization error under the assumption that the score function is perfectly estimated (i.e., $N, M \to \infty$). Each step simulates the time evolution over an interval of length $\eta := h/T$, corresponding to the segment $(\ell - 1)\eta + t \in [(\ell - 1)\eta, \ell\eta]$ with $t \in [0, \eta]$ for $\ell = 1, \dots, T$. In the ideal dynamics (5), the drift term is time-dependent, whereas in the discretized scheme we approximate it by fixing the drift at the beginning of each step. Following an argument similar to that in Vempala & Wibisono (2019), we obtain the following result.

**Lemma 3.** *Fix a step index $\ell \in \{1, \dots, T\}$ and let $\eta := h/T$. Let $(\nu_t)_{t \in [0,\eta]}$ denote the ideal backward marginals in (5) at elapsed time $(\ell - 1)\eta + t$ for $t \in [0, \eta]$, and $(\mu_t)_{t \in [0,\eta]}$ denote the frozen-drift approximation within this step. Then for $u \geq 1$ and all $t \in [0, \eta]$,*

$$\frac{\mathrm{d}}{\mathrm{d}t} H_{\nu_t}(\mu_t) \leq -\left(\frac{1}{2} - \frac{1}{4u^2}\right) J_{\nu_t}(\mu_t) + u^2 \, \mathbb{E}_{\mu_{0,t}}\big[\|\tilde{s}_0(z_0) - \tilde{s}_t(z_t)\|^2\big]. \tag{16}$$

*Equivalently, integrating over $t \in [0, \eta]$ yields*

$$H_{\nu_\eta}(\mu_\eta) - H_{\nu_0}(\mu_0) \leq -\left(\frac{1}{2} - \frac{1}{4u^2}\right) \int_0^\eta J_{\nu_t}(\mu_t) \, \mathrm{d}t + u^2 \int_0^\eta \mathbb{E}_{\mu_{0,t}}\big[\|\tilde{s}_0(z_0) - \tilde{s}_t(z_t)\|^2\big] \, \mathrm{d}t.$$

*Proof.* For convenience to analyze the behavior at $(\ell - 1)\eta + t$ with $t \in [0, \eta]$ which corresponds to the time interval $[(\ell - 1)\eta, \ell\eta]$ in original backward process defined in (5), we restate the SDE and the associated Fokker–Planck equation as: for $t \in [0, \eta]$,

$$\mathrm{d}Z_t^\leftarrow = \tilde{s}_t(Z_t^\leftarrow) \, \mathrm{d}t + \mathrm{d}B_t, \quad \mathrm{law}(Z_t^\leftarrow) = \nu_t,$$

$$\partial_t \nu_t = -\mathrm{div}(\nu_t \tilde{s}_t) + \frac{1}{2}\Delta\nu_t = \nabla \cdot \left(\nu_t\big(-\tilde{s}_t + \frac{1}{2}\nabla \log \nu_t\big)\right),$$

where we define the score function $\tilde{s}_t := \nabla \log\big(\pi^X P_{h-(\ell-1)\eta-t}\big)$.

In contrast, the time-discretized approximation corresponds to the process

$$\mathrm{d}z_t = \tilde{s}_0(z_0) \, \mathrm{d}t + \mathrm{d}B_t, \quad \mathrm{law}(z_t) = \mu_t, \tag{17}$$

where the drift is frozen at the beginning of the step.

The associated Fokker–Planck equation is

$$\partial_t \mu_t(z_t \mid z_0) = -\mathrm{div}\big(\mu_t(z_t \mid z_0) \, \tilde{s}_0(z_0)\big) + \frac{1}{2}\Delta\mu_t(z_t \mid z_0)$$

$$= \nabla \cdot (\mu_t(z_t \mid z_0)(-\tilde{s}_0(z_0) + \frac{1}{2}\nabla \log \mu_t(z_t \mid z_0))).$$

Averaging over $z_0 \sim \mu_0$, this becomes

$$\partial_t \mu_t = \nabla \cdot \left(\mu_t\big(-\mathbb{E}_{\mu_{0|t}}[\tilde{s}_0(z_0) \mid z_t] + \frac{1}{2}\nabla \log \mu_t\big)\right).$$

The time derivative of the KL divergence is then

$$\frac{\mathrm{d}}{\mathrm{d}t} H_{\nu_t}(\mu_t) = \frac{\mathrm{d}}{\mathrm{d}t} \int \mu_t \log \frac{\mu_t}{\nu_t} \, \mathrm{d}z$$

$$= \int \left[(\partial_t \mu_t) \log \frac{\mu_t}{\nu_t} + \left(\partial_t \mu_t - \mu_t \frac{\partial_t \nu_t}{\nu_t}\right)\right] \mathrm{d}z$$

$$= \int \left[\nabla \cdot \left(\mu_t\big(-\mathbb{E}_{\mu_{0|t}}[\tilde{s}_0(z_0) \mid z_t] + \frac{1}{2}\nabla \log \mu_t\big)\right) \log \frac{\mu_t}{\nu_t}\right.$$

$$\left. - \frac{\mu_t}{\nu_t} \nabla \cdot \left(\nu_t\big(-\tilde{s}_t + \frac{1}{2}\nabla \log \nu_t\big)\right)\right] \mathrm{d}z$$

$$= \int \left[-\left\langle \mu_t\big(-\mathbb{E}_{\mu_{0|t}}[\tilde{s}_0(z_0) \mid z_t] + \frac{1}{2}\nabla \log \mu_t\big), \nabla \log \frac{\mu_t}{\nu_t}\right\rangle\right.$$

$$\left. + \left\langle \nu_t\big(-\tilde{s}_t + \frac{1}{2}\nabla \log \nu_t\big), \frac{\mu_t}{\nu_t}\nabla \log \frac{\mu_t}{\nu_t}\right\rangle\right] \mathrm{d}z$$

$$= \int \left\langle -\frac{1}{2}\nabla \log \frac{\mu_t}{\nu_t} + \mathbb{E}_{\mu_{0|t}}[\tilde{s}_0(z_0) \mid z_t] - \tilde{s}_t, \nabla \log \frac{\mu_t}{\nu_t}\right\rangle \mu_t \, \mathrm{d}z.$$

Here, the third equality substitutes the Fokker–Planck equations for $\partial_t \mu_t$ and $\partial_t \nu_t$ and also uses that the integral of $\partial_t \mu_t$ vanishes due to mass conservation. The fourth equality uses integration by parts (assuming sufficiently fast decay at infinity) and the identity.

Simplifying, we obtain

$$\frac{\mathrm{d}}{\mathrm{d}t} H_{\nu_t}(\mu_t) = -\frac{1}{2} J_{\nu_t}(\mu_t) + \mathbb{E}_{\mu_t}\left[\langle \mathbb{E}_{\mu_{0|t}}[\tilde{s}_0(z_0) \mid z_t] - \tilde{s}_t, \nabla \log \frac{\mu_t}{\nu_t}\rangle\right].$$

Applying $\langle a, b\rangle \leq u^2 \|a\|^2 + \frac{1}{4u^2}\|b\|^2$, we conclude

$$\frac{\mathrm{d}}{\mathrm{d}t} H_{\nu_t}(\mu_t) \leq -\left(\frac{1}{2} - \frac{1}{4u^2}\right) J_{\nu_t}(\mu_t) + u^2 \mathbb{E}_{\mu_{0,t}(z_0, z_t)}\left[\|\tilde{s}_0(z_0) - \tilde{s}_t(z_t)\|^2\right].$$

$\square$

This establishes the discretization error bound of one diffusion step in terms of the deviation between the frozen score $\tilde{s}_0$ and the ideal time-dependent score $\tilde{s}_t$.

**Assumption 1** (Smoothness of the interim distribution). *The interim distribution $\pi^X P_{h-(\ell-1)\eta}$ is $L_\ell$-smooth, i.e., the potential gradient $\nabla \log(\pi^X P_{h-(\ell-1)\eta})$ is $L_\ell$-Lipschitz.*

**Assumption 2** (Lipschitz condition along the time direction). *The expected score satisfies a Lipschitz-type condition along the time direction:*

$$\mathbb{E}_{\mu_t}\left[\|\tilde{s}_0(z_t) - \tilde{s}_t(z_t)\|^2\right] \leq C_{t,\ell}^2\, t^2.$$

**Corollary 1.** *Assume $\nu_0 := \pi^X P_{(\ell-1)\eta}$ satisfies LSI with constant $C_{LSI}(\nu_0)$ and under Assumption 1 and Assumption 2,*

$$\frac{\mathrm{d}}{\mathrm{d}t} H_{\nu_t}(\mu_t) \leq -\left(\frac{1}{2} - \frac{1}{4u^2}\right) J_{\nu_t}(\mu_t) + u^2\left(4\eta^2 L_\ell^4\, C_{LSI}(\nu_0)\, H_{\nu_0}(\mu_0) + \eta d C\right), \quad (18)$$

$$\text{where} \quad C = \sup_l \sup_t 2t L_\ell^3 + L_\ell^2 + t C_{t,\ell}^2/d. \quad (19)$$

*Proof.* We bound the second term of the right hand side in the inequality in Lemma 3 using Assumption 1 and Assumption 2,

$$\mathbb{E}_{\mu_{0,t}(z_0, z_t)}\left[\|\tilde{s}_0(z_0) - \tilde{s}_t(z_t)\|^2\right] \leq \mathbb{E}_{\mu_{0,t}(z_0, z_t)}\left[\|\tilde{s}_0(z_0) - \tilde{s}_0(z_t)\|^2\right] + \mathbb{E}_{\mu_t}\left[\|\tilde{s}_0(z_t) - \tilde{s}_t(z_t)\|^2\right]$$
$$\leq L_\ell^2 \mathbb{E}_{\mu_{0,t}(z_0, z_t)}\left[\|z_0 - z_t\|^2\right] + C_{t,\ell}^2\, t^2.$$

Under the discretization update

$$z_t = z_0 + \tilde{s}_0(x_0)\, t + \sqrt{t}\, \xi, \quad \xi \sim \mathcal{N}(0, I_d),$$

we have

$$\mathbb{E}_{\mu_{0,t}(z_0, z_t)}\left[\|z_0 - z_t\|^2\right] = \mathbb{E}_{\mu_0}[\|\tilde{s}_0(z_0)\, t + \sqrt{t}\, \xi\|^2]$$
$$= t^2\, \mathbb{E}_{\mu_0}[\|\tilde{s}_0(z_0)\|^2] + td$$
$$\leq t^2(4L_\ell^2\, C_{LSI}(\nu_0)\, H_{\nu_0}(\mu_0) + 2dL_\ell) + td.$$

The last inequality comes from Lemma 12 in Vempala & Wibisono (2019) with Assumption 1 and $\nu_0$ satisfying Talagrand's inequality with constant $C_{LSI}(\nu_0)^{-1}$. Putting them all together, we obtain (18) where $C$ is independent of $\ell$ by taking the supremum as (19). $\square$

**Proposition 2** (One-step bound for the diffusion-approximated proximal sampler without score estimation error). *Assume Assumption 1 and Assumption 2 hold, and that $\pi^X$ satisfies an LSI with constant $C_{LSI}(\pi^X)$. Let $C$ be as in (19). Let overall step size $h > 0$ be split into $T$ steps with $\eta := h/T$. In the regime $N, M \to \infty$ where the score estimation error vanishes, the distribution at the $k$-th iteration $\rho_k^X$ satisfies*

$$H_{\pi^X}(\rho_{k+1}^X) \leq \frac{H_{\pi^X}(\rho_k^X)}{(1 + h/C_{LSI}(\pi^X))^{2-1/(2u^2)}}$$
$$+ 2u^4 C_{LSI}(\pi^X)\left((1 + h/C_{LSI}(\pi^X))^{1/(2u^2)} - 1\right)\left\{4\eta^2 \bar{L}^4(C_{LSI}(\pi^X) + h)\bar{H} + \eta d C\right\}$$

*for $u \geq 1$, where $\bar{H}$ is the supremum of KL divergence between the interim distribution of the backward denoised distribution at timestep $t = \ell\eta$ and $\pi^Y Q_{\ell\eta}$ and $\bar{L} = \sup_\ell L_\ell$*

*Proof.* As shown in Corollary 13 of Chafaï (2004), the logarithmic Sobolev constant satisfies

$$C_{LSI}(\pi^X P_t) \le C_{LSI}(\pi^X) + t.$$

For $t \in [0, \eta]$, the law $\nu_t$ corresponds to the ideal denoising path at time $(\ell - 1)\eta + t$, namely $\nu_t = \pi^X P_{h-(\ell-1)\eta-t}$. Hence its logarithmic Sobolev constant can be bounded as

$$C_{LSI}(\nu_t) \le C_{LSI}(\pi^X) + h - (\ell - 1)\eta - t =: \alpha_\ell - t.$$

From Corollary 1 combined with LSI, we obtain

$$\frac{\mathrm{d}}{\mathrm{d}t} H_{\nu_t}(\mu_t) \le -\frac{1 - 1/(2u^2)}{\alpha_\ell - t} H_{\nu_t}(\mu_t) + u^2 [4\eta^2 L_\ell^4 C_{LSI}(\nu_0) H_{\nu_0}(\mu_0) + \eta dC].$$

Applying Gronwall's inequality, where we multiply by the integrating factor: $(\alpha_\ell - t)^{-1+1/(2u^2)}$,

$$\frac{\mathrm{d}}{\mathrm{d}t} \left\{ (\alpha_\ell - t)^{-1+1/(2u^2)} H_{\nu_t}(\mu_t) \right\} \le (\alpha_\ell - t)^{-1+1/(2u^2)} u^2 \left[ 4\eta^2 L_\ell^4 C_{LSI}(\nu_0) H_{\nu_0}(\mu_0) + \eta dC \right]$$

and integrating over $t \in [0, \eta]$, we obtain

$$(\alpha_\ell - \eta)^{-1+1/(2u^2)} H_{\nu_\eta}(\mu_\eta) - \alpha_\ell^{-1+1/(2u^2)} H_{\nu_0}(\mu_0)$$
$$\le 2u^4 (\alpha_\ell^{1/(2u^2)} - (\alpha_\ell - \eta)^{1/(2u^2)}) \left\{ 4\eta^2 L_\ell^4 C_{LSI}(\nu_0) H_{\nu_0}(\mu_0) + \eta dC \right\}.$$

Recall that $\mu_t, \nu_t$ are defined in the $\ell$-th step of the time-discretized diffusion. Using the fact that $\mu_\eta, \nu_\eta$ are equivalent to $\mu_0, \nu_0$ in the $(\ell + 1)$-th step and $\alpha_\ell - \eta = \alpha_{\ell+1}$, we iterate this inequality over $\ell = 1, \dots, T$ to obtain

$$C_{\mathrm{LSI}}(\pi^X)^{-1+1/(2u^2)} H_{\pi^X}(\rho_{k+1}^X) - (C_{\mathrm{LSI}}(\pi^X) + h)^{-1+1/(2u^2)} H_{\pi^Y}(\rho_{k+1/2}^Y)$$
$$\le 2u^4 \left( (C_{\mathrm{LSI}}(\pi^X) + h)^{1/(2u^2)} - C_{\mathrm{LSI}}(\pi^X)^{1/(2u^2)} \right) \left\{ 4\eta^2 \bar{L}^4 (C_{\mathrm{LSI}}(\pi^X) + h)\bar{H} + \eta dC \right\},$$

where $\bar{H} := \sup_l H_{\nu_0}(\mu_0)$, the supremum of the KL divergence between the updated distribution in each diffusion step and the corresponding ideal distribution without discretization error and $\bar{L} := \sup_\ell L_\ell$.

Equivalently,

$$H_{\pi^X}(\rho_{k+1}^X) \le \frac{H_{\pi^Y}(\rho_{k+1/2}^Y)}{(1 + h/C_{\mathrm{LSI}}(\pi^X))^{1-1/(2u^2)}}$$
$$+ 2u^4 C_{\mathrm{LSI}}(\pi^X) \left( (1 + h/C_{LSI}(\pi^X))^{1/(2u^2)} - 1 \right) \left\{ 4\eta^2 \bar{L}^4 (C_{LSI}(\pi^X) + h)\bar{H} + \eta dC \right\}.$$

This corresponds to the time-discretized version of the KL contraction in (8) with $q = 1$. We complete the proof by applying inequality (7) in Lemma 1 with $q = 1$.

$\square$

A.4 ANALYSIS FOR SCORE ESTIMATION ERROR

In this section, we evaluate the error of score estimation under a set of assumptions. In particular, we prove that the estimation error scales as $O(1/N + 1/M)$, which converges to zero in the limit of $N, M \to \infty$.

Recall that $\hat{q}_{k+1/2}(y \mid X_k) = \frac{1}{N} \sum_i \mathcal{N}(y; x_i, hI_d)$ and we further define (unnormalized) Gaussian mixture density

$$\hat{g}_N^{k+\frac{1}{2}}(x \mid U) := \frac{1}{N} \sum_{j=1}^{N} \frac{\mathcal{N}(x; y_j, hI_d)}{\hat{q}_{k+\frac{1}{2}}(y_j \mid X_k)},$$

which is conditioned on $U = (Y_{k+1/2}, X_k)$ s.t. $X_k \sim_N \rho_k^X$, $Y_{k+1/2} \sim_N \hat{q}_{k+1/2}(\cdot | X_k)$.

At time $t$, the true score in the surrogate dynamics is expressed as

$$\hat{s}_N(z_t, t) = \nabla \log\big((\hat{g}_N^{k+\frac{1}{2}} \mid_U e^{-f})P_{h-t}\big)(z_t).$$

We approximate this by

$$\hat{s}_{N,M}(z_t, t) = \frac{\frac{1}{M} \sum_{m=1}^{M} \exp\big(-f(\hat{z}^{(m)})\big)(\hat{z}^{(m)} - z_t)/(h-t)}{\frac{1}{M} \sum_{m=1}^{M} \exp\big(-f(\hat{z}^{(m)})\big)}, \quad \hat{z}^{(m)} \sim \hat{g}_N^{k+\frac{1}{2}}|_U(\cdot \mid z_t). \quad (20)$$

Compared to the setting with only time discretization in Section A.3, the drift term $\tilde{s}_0$ in one denoising step of the discretized dynamics (17) is replaced by the estimator $\hat{s}_{N,M}(\cdot, (\ell-1)\eta)$ given in (20). Using this substitution, the statement of Lemma 3, originally expressed as (16), is modified accordingly as follows:

$$\frac{\mathrm{d}}{\mathrm{d}t} H_{\nu_t}(\mu_t) \leq -\Big(\frac{1}{2} - \frac{1}{4u^2}\Big) J_{\nu_t}(\mu_t) + u^2 \mathbb{E}_{\mu_{0,t}}\big[\|\hat{s}_{N,M}(z_0, (\ell-1)\eta) - \tilde{s}_t(z_t)\|^2 \mid U, \hat{Z}|_U\big]$$

where $U \sim \hat{q}_{k+1/2,k}$ and $\hat{Z}|_U \sim_M \hat{g}_N^{k+1/2}|_U(\cdot \mid z_0)$.

The expectation in the second term on the right-hand side is evaluated as:

$$\mathbb{E}_{\mu_{0,t}}\big[\|\hat{s}_{N,M}(z_0, (\ell-1)\eta) - \tilde{s}_t(z_t)\|^2 \mid U, \hat{Z}|_U\big]$$
$$\leq 2\mathbb{E}_{\mu_{0,t}}\big[\|\tilde{s}_0(z_0) - \tilde{s}_t(z_t)\|^2\big] + 2\mathbb{E}_{\mu_0}\big[\|\hat{s}_{N,M}(z_0, (\ell-1)\eta) - \tilde{s}_0(z_0)\|^2 \mid U, \hat{Z}|_U\big]$$
$$\leq 2\mathbb{E}_{\mu_{0,t}}\big[\|\tilde{s}_0(z_0) - \tilde{s}_t(z_t)\|^2\big] + 4\mathbb{E}_{\mu_0}\big[\|\hat{s}_N(z_0, (\ell-1)\eta) - \tilde{s}_0(z_0)\|^2 \mid U\big]$$
$$+ 4\mathbb{E}_{\mu_0}\big[\|\hat{s}_{N,M}(z_0, (\ell-1)\eta) - \hat{s}_N(z_0, (\ell-1)\eta)\|^2 \mid U, \hat{Z}|_U\big], \quad (21)$$

which means we can independently evaluate the errors from time discretization (the first term), from finite $N$ (the second term) and from finite $M$ (the third term).

**Score error from finite $N$.** Let $\tau := h - (\ell-1)\eta$, so that $z_\tau$ coincides with $z_0$, the initial sample of the $\ell$-th denoising step. For $z_0 \in \mathbb{R}^d$, define the conditional law, where denoised $\hat{z}$ is conditioned on $z_\tau$,

$$\pi^X(\hat{z} \mid z_\tau) \propto \exp\Big(-f(\hat{z}) - \frac{\|\hat{z} - z_\tau\|^2}{2\tau}\Big),$$

with normalizer

$$Z(z_\tau) := \int \exp\Big(-f(\hat{z}) - \frac{\|\hat{z} - z_\tau\|^2}{2\tau}\Big) \mathrm{d}\hat{z}.$$

Let

$$G(z_\tau \mid U) := \frac{1}{Z(z_\tau)} \int \hat{g}_N^{k+\frac{1}{2}}(\hat{z} \mid U) \exp\Big(-f(\hat{z}) - \frac{\|\hat{z} - z_\tau\|^2}{2\tau}\Big) \mathrm{d}\hat{z}$$
$$= \int \hat{g}_N^{k+\frac{1}{2}}(\hat{z} \mid U) \pi^X(\hat{z} \mid z_\tau) \, d\hat{z}.$$

Here we assume:

**Assumption 3** (Bounded Covariance of the conditioned target distribution). *For all $\ell = 1, \cdots, T$ and $z_\tau$, there exists $C_V > 0$ s.t. $\sup \operatorname{Var}_{\pi^X(\hat{z}|z_\tau)}[\hat{z}] \le C_V$.*

**Assumption 4** (Lower bound of density ratio between the perturbed distributions).

$$\frac{\pi^X(\hat{z} \mid z_\tau) * \mathcal{N}(0, hI_d)(u)}{\hat{q}_{k+1/2}(u \mid X_k)} \ge \kappa > 0$$

*for a.e. $u$, hence $G(z_\tau \mid U) \ge \kappa$.*

**Lemma 4** (finite-$N$ error is $O(1/N)$). *Under Assumption 3 and Assumption 4, the expectation of the error term induced by the finite base sample size $N$ satisfies*

$$\mathbb{E}_{\mu_0, \hat{q}_{k+1/2}}\Big[\|\hat{s}_N(z_\tau, h - \tau) - s_{h-\tau}(z_\tau)\|^2\Big]$$

$$\le \frac{2\kappa^{-2}C_V}{N\tau^2} \mathbb{E}_{\mu_0, \rho_k^X, \pi^X(\hat{z}|z_\tau)}\Big[\chi^2\big(\mathcal{N}(\hat{z}, hI_d) \,\|\, \hat{q}_{k+\frac{1}{2}}|X_k\big) + \chi^2\big(\pi^X(\hat{z}|z_\tau) * \mathcal{N}(0, hI_d) \,\|\, \hat{q}_{k+\frac{1}{2}}|X_k\big)\Big].$$

*In particular, under bounded $\chi^2$-divergences, the second term on the right-hand side of the inequality (21) is $O(1/N)$.*

*Proof.* By Tweedie's formula at time $\tau = h - (\ell - 1)\eta$, we have

$$\mathbb{E}_{\mu_0}\Big[\|\hat{s}_N(z_\tau, h - \tau) - s_{h-\tau}(z_\tau)\|^2 \,\Big|\, U\Big] = \frac{1}{\tau^2} \mathbb{E}_{\mu_0}\left[\left\|\mathbb{E}_{\hat{z}\sim\hat{g}_N^{k+\frac{1}{2}}|_U(\cdot|z_\tau)}[\hat{z}] - \mathbb{E}_{\hat{z}\sim\pi^X(\cdot|z_\tau)}[\hat{z}]\right\|^2\right]$$

$$= \frac{1}{\tau^2} \mathbb{E}_{\mu_0}\left[\left\|\int \hat{z}\Big(\frac{\hat{g}_N^{k+\frac{1}{2}}(\hat{z} \mid U)}{G(z_\tau \mid U)} - 1\Big) \pi^X(\hat{z} \mid z_\tau) \, d\hat{z}\right\|^2\right].$$

Here we used that $\hat{g}_N^{k+\frac{1}{2}}|_U(\hat{z} \mid z_\tau) \propto \hat{g}_N^{k+\frac{1}{2}}(\hat{z} \mid U) \exp\Big(-f(\hat{z}) - \frac{\|\hat{z}-z_\tau\|^2}{2\tau}\Big)$, and substituted the definition of $G(z_\tau \mid U)$ to obtain the normalized form.

Let $\bar{c}(z_\tau) := \int \hat{z} \, \pi^X(\hat{z} \mid z_\tau) \, d\hat{z}$. Since $\int (\frac{\hat{g}_N^{k+1/2}(\hat{z}|U)}{G(z_\tau|U)} - 1) \, \pi^X(\hat{z} \mid z_\tau) \, d\hat{z} = 0$, we can center the integrand and apply Cauchy–Schwarz to obtain

$$\mathbb{E}_{\mu_0}\Big[\|\hat{s}_N(z_\tau, h - \tau) - s_{h-\tau}(z_\tau)\|^2 \,\Big|\, U\Big]$$

$$= \frac{1}{\tau^2} \mathbb{E}_{\mu_0}\left[\left\|\int \big(\hat{z} - \bar{c}(z_\tau)\big)\Big(\frac{\hat{g}_N^{k+\frac{1}{2}}(\hat{z} \mid U)}{G(z_\tau \mid U)} - 1\Big) \pi^X(\hat{z} \mid z_\tau) \, d\hat{z}\right\|^2\right]$$

$$\le \frac{1}{\tau^2} \mathbb{E}_{\mu_0}\Big[\operatorname{Var}_{\pi^X(\hat{z}|z_\tau)}[\hat{z}] \cdot \operatorname{Var}_{\pi^X(\hat{z}|z_\tau)}\big[\frac{\hat{g}_N^{k+\frac{1}{2}}(\hat{z} \mid U)}{G(z_\tau \mid U)}\big]\Big]$$

$$\le \frac{C_V}{\tau^2} \mathbb{E}_{\mu_0}\left[\int \frac{1}{G(z_\tau \mid U)^2} \big(\hat{g}_N^{k+\frac{1}{2}}(\hat{z} \mid U) - G(z_\tau \mid U)\big)^2 \pi^X(\hat{z} \mid z_\tau) \, d\hat{z}\right]$$

$$\le \frac{\kappa^{-2}C_V}{\tau^2} \mathbb{E}_{\mu_0}\left[\int \big(\hat{g}_N^{k+\frac{1}{2}}(\hat{z} \mid U) - G(z_\tau \mid U)\big)^2 \pi^X(\hat{z} \mid z_\tau) \, d\hat{z}\right].$$

The second inequality is from Assumption 3, and Assumption 4 yields the last inequality. Using the unbiasedness identities with $u_j \sim \hat{q}_{k+1/2}(\cdot \mid X_k)$,

$$\mathbb{E}_{u_j\sim\hat{q}_{k+1/2}(\cdot|X_k)}\left[\frac{\mathcal{N}(\hat{z}; u_j, hI_d)}{\hat{q}_{k+1/2}(u_j \mid X_k)}\right] = 1, \quad \mathbb{E}_{u_j\sim\hat{q}_{k+1/2}(\cdot|X_k)}\left[\frac{\pi^X(\hat{z} \mid z_\tau) * \mathcal{N}(0, hI_d)(u_j)}{\hat{q}_{k+1/2}(u_j \mid X_k)}\right] = 1,$$

and the decomposition $\big((\hat{g} - 1) + (1 - G)\big)^2 \le 2(\hat{g} - 1)^2 + 2(G - 1)^2$, we get

$$\mathbb{E}_{\mu_0, \hat{q}_{k+1/2}}\Big[\|\hat{s}_N(z_\tau, h - \tau) - s_{h-\tau}(z_\tau)\|^2\Big]$$

$$\le \frac{2\kappa^{-2}C_V}{\tau^2} \mathbb{E}_{\mu_0, \hat{q}_{k+1/2}}\left[\int \Big((\hat{g}_N^{k+\frac{1}{2}}(\hat{z} \mid U) - 1)^2 + (G(z_\tau \mid U) - 1)^2\Big) \pi^X(\hat{z} \mid z_\tau) \, d\hat{z}\right].$$

Standard variance calculations for importance-weighted kernel mixtures yield

$$\mathbb{E}\left[\int (\hat{g}_N^{k+\frac{1}{2}}(\hat{z} \mid U) - 1)^2 \, \pi^X(\hat{z} \mid z_\tau) \, \mathrm{d}\hat{z}\right] = \mathbb{E}_{\mu_0, \rho_k^X, \pi^X(\hat{z}|z_\tau)}\left[\frac{1}{N}\mathrm{Var}_{\hat{q}_{k+1/2}|X_k}\left[\frac{\mathcal{N}(\hat{z}; u_j, hI_d)}{\hat{q}_{k+1/2}(u_j|X_k)}\right]\right],$$

$$\mathbb{E}\left[\int (G(z_\tau) - 1)^2 \, \pi^X(\hat{z} \mid z_\tau) \, \mathrm{d}\hat{z}\right] = \mathbb{E}_{\mu_0, \rho_k^X, \pi^X(\hat{z}|z_\tau)}\left[\frac{1}{N}\mathrm{Var}_{\hat{q}_{k+1/2}|X_k}\left[\frac{\pi^X(\hat{z}|z_\tau)*\mathcal{N}(0,hI_d)(u_j)}{\hat{q}_{k+1/2}(u_j|X_k)}\right]\right].$$

Each variance is upper bounded by the corresponding $\chi^2$-divergence, yielding the claim. $\qquad\square$

**Score estimation error from finite $M$.** Here we assume

**Assumption 5.** *The following fourth moments under $\hat{g}_N^{k+\frac{1}{2}}|_U(\cdot \mid z_\tau)$ are finite:*

$$\mathbb{E}\left[\exp\big(4f(\hat{z})\big)\right] < \infty, \quad \mathbb{E}\left[\|\hat{z} - z_\tau\|^4 \exp\big(4f(\hat{z})\big)\right] < \infty.$$

This can be satisfied, for instance, when the variance of each component in the Gaussian mixture $\hat{g}_N^{k+1/2}|_U$ is sufficiently small compared to the growth rate of $f$.

**Lemma 5** (finite-$M$ error is $O(1/M)$)**.** *Fix $z_\tau$ and $U \sim_N \hat{q}_{k+\frac{1}{2}}(\cdot \mid X_k)$. Under Assumption 5, there exists a constant $C_M$ such that*

$$\mathbb{E}_{\mu_0, \hat{g}_N^{k+1/2}|_U}\big[\|\hat{s}_{N,M}(z_\tau, h - \tau) - \hat{s}_N(z_\tau, h - \tau)\|^2 \,\big|\, U\big] \leq \frac{C_M}{M\,\tau^2}.$$

*Proof.* Write the estimator (20) at time $\tau$ as a ratio of empirical means as follows:

$$\hat{s}_{N,M}(z_\tau, \tau) = \frac{A_M}{\tau B_M}, \quad A_M := \frac{1}{M}\sum_{m=1}^M (\hat{z}^{(m)} - z_\tau)\, e^{-f(\hat{z}^{(m)})}, \quad B_M := \frac{1}{M}\sum_{m=1}^M e^{-f(\hat{z}^{(m)})}.$$

Let $A := \mathbb{E}[A_M]$ and $B := \mathbb{E}[B_M]$ under $\hat{z}^{(m)} \overset{\text{i.i.d.}}{\sim} \hat{g}_N^{k+\frac{1}{2}}|_U(\cdot \mid z_\tau)$. Then $\hat{s}_N(z_\tau, h - \tau) = A/(\tau B)$ and

$$\|\hat{s}_{N,M} - \hat{s}_N\|^2 = \frac{1}{\tau^2}\left\|\frac{A_M}{B_M} - \frac{A}{B}\right\|^2 = \frac{1}{\tau^2}\left\|\frac{A_M B - A B_M}{B_M B}\right\|^2.$$

Use the decomposition $A_M B - A B_M = (A_M - A)B + A(B - AB_M)$ to obtain the crude bound

$$\left\|\frac{A_M}{B_M} - \frac{A}{B}\right\|^2 \leq \frac{2\|A_M - A\|^2}{B_M^2} + \frac{2\|A\|^2 \|B_M - B\|^2}{B_M^2 B^2}.$$

Since $e^{-f(\hat{z})} > 0$, Jensen implies $B_M^{-2} \leq \frac{1}{M}\sum_{m=1}^M e^{2f(\hat{z}^{(m)})}$. Hence

$$\|\hat{s}_{N,M} - \hat{s}_N\|^2$$
$$\leq \frac{2}{\tau^2}\left\{\left(\frac{1}{M}\sum_{m=1}^M e^{2f(\hat{z}^{(m)})}\right)\|A_M - A\|^2 + \frac{\|A\|^2}{B^2}\left(\frac{1}{M}\sum_{m=1}^M e^{2f(\hat{z}^{(m)})}\right)\|B_M - B\|^2\right\}.$$

Taking conditional expectation with respect to $\hat{Z}|_U \sim \hat{g}_N^{k+1/2}|_U$ and applying Hölder's inequality with exponents $(2, 2)$ to each term, we have

$$\mathbb{E}\big[\|\hat{s}_{N,M} - \hat{s}_N\|^2 \,\big|\, U\big]$$
$$\leq \frac{2}{\tau^2}\left\{(\mathbb{E}[X_M^2])^{\frac{1}{2}}\big(\mathbb{E}[\|A_M - A\|^4]\big)^{\frac{1}{2}} + \frac{\|A\|^2}{B^2}(\mathbb{E}[X_M^2])^{\frac{1}{2}}\big(\mathbb{E}[(B_M - B)^4]\big)^{\frac{1}{2}}\right\},$$

where $X_M := \frac{1}{M}\sum_{m=1}^M e^{2f(\hat{z}^{(m)})}$. We compute the second moment of $X_M$ explicitly:

$$\mathbb{E}\big[X_M^2\big] = \frac{1}{M^2}\mathbb{E}\left[\sum_m (e^{2f(\hat{z}^{(m)})})^2 + \sum_{m \neq m'} e^{2f(\hat{z}^{(m)})} e^{2f(\hat{z}^{(m')})}\right]$$
$$= \frac{1}{M}\mathbb{E}\left[e^{4f(\hat{z})}\right] + \frac{M-1}{M}\big(\mathbb{E}[e^{2f(\hat{z})}]\big)^2.$$

To analyze the fourth moment of $\sum_{m=1}^{M} v_m$ where $\mathbb{E}[v_m] = 0$, note that

$$\left\| \sum_{m=1}^{M} v_m \right\|^4 = \left\langle \sum_i v_i, \sum_j v_j \right\rangle^2 = \left( \sum_{i,j} \langle v_i, v_j \rangle \right)^2 = \sum_{i,j,k,l} \langle v_i, v_j \rangle \langle v_k, v_l \rangle.$$

Taking expectations, only the terms in which each index appears an even number of times contribute, and hence

$$\mathbb{E} \left\| \sum_{m=1}^{M} v_m \right\|^4 = M \, \mathbb{E}\|v_1\|^4 + M(M-1)\big(\mathbb{E}\|v_1\|^2\big)^2 + 2M(M-1)\mathbb{E}\big|\langle v_1, v_2 \rangle\big|^2$$

$$\leq M \, \mathbb{E}\|v_1\|^4 + 3M(M-1)\big(\mathbb{E}\|v_1\|^2\big)^2.$$

Thus, we obtain

$$\mathbb{E}\left[ \|A_M - A\|^4 \right]$$
$$\leq \frac{1}{M^3} \mathbb{E}\left[ \left\| ((\hat{z} - z_\tau)e^{-f(\hat{z})} - \mathbb{E}[(\hat{z} - z_\tau)e^{-f(\hat{z})}]) \right\|^4 \right] + \frac{3}{M^2} \operatorname{Var}\left[ (\hat{z} - z_\tau)e^{-f(\hat{z})} \right]^2,$$
$$\mathbb{E}\left[ (B_M - B)^4 \right] \leq \frac{1}{M^3} \mathbb{E}\left[ \left( e^{-f(\hat{z})} - \mathbb{E}[e^{-f(\hat{z})}] \right)^4 \right] + \frac{3}{M^2} \operatorname{Var}\left[ e^{-f(\hat{z})} \right]^2.$$

Substituting these bounds back gives

$$\mathbb{E}\left[\|\hat{s}_{N,M} - \hat{s}_N\|^2 \,\middle|\, U\right] \leq \frac{2}{\tau^2} \left( \mathbb{E}[X_M^2] \right)^{\frac{1}{2}} \left\{ \left( \frac{1}{M^3} \mathbb{E}\big[\|\cdot\|^4\big] + \frac{3}{M^2} \operatorname{Var}\left[ (\hat{z} - z_\tau)e^{-f(\hat{z})} \right]^2 \right)^{\frac{1}{2}} \right.$$
$$\left. + \frac{\|A\|^2}{B^2} \left( \frac{1}{M^3} \mathbb{E}\left[ \left( e^{-f(\hat{z})} - \mathbb{E}[e^{-f(\hat{z})}] \right)^4 \right] + \frac{3}{M^2} \operatorname{Var}\left[ e^{-f(\hat{z})} \right]^2 \right)^{\frac{1}{2}} \right\}.$$

If the fourth moments are finite under Assumption 5, $\mathbb{E}[X_M^2] = O(1)$ and each rooted bracket is $O(M^{-1})$. Therefore,

$$\mathbb{E}\left[\|\hat{s}_{N,M}(z_\tau, \tau) - \hat{s}_N(z_\tau, \tau)\|^2 \,\middle|\, U\right] \leq \frac{C_M}{M \, \tau^2},$$

for a constant $C_M$ depending only on the above moments. $\qquad\square$

Combining Lemma 4 and Lemma 5, we conclude the overall score estimation error is $O(1/N + 1/M)$ under several appropriate assumptions.

# B    EXPERIMENT DETAILS

We provide parameters of each method for our experiments in Table 1 and Table 2. All experiments were conducted on an Intel Xeon CPU Max 9480 without GPU acceleration.

**KL divergence estimation for the Gaussian Lasso experiment.**    Following the setting of Liang & Chen (2023b), we first ran the proximal sampler for 100,000 burn-in iterations, and then continued for 400,000 iterations. From this long trajectory we randomly collected 1000 samples to serve as reference particles. At each evaluation of the experiment, the KL divergence was estimated using 1000 current particles and these 1000 reference particles. For estimation we used the $k$-nearest-neighbor estimator (Kozachenko & Leonenko, 1987) implemented in Büth et al. (2025), with $k = 4$ as recommended by Kraskov et al. (2004). Although this estimation involves sampling error, we repeated the entire experiment with 10 different random seeds and, for each seed, computed the KL divergence based on the corresponding set of particles. We then reported the mean and variance of these estimates across the 10 runs to provide a more robust evaluation.

Table 1: Parameter setting for the experiment in Section 5.1.

| Method | Parameters |
|---|---|
| Proximal Sampler with RGO | Initial distribution: $y_{1/2}^{(j)} \sim \mathcal{N}(0, I_d)$
Step size: $\eta = 1/135$
Number of independent chains: 100
Thinning: 10 |
| Ours | Initial distribution: $x_0^{(i)} \sim \mathcal{N}(0, I_d)$
Step size: $h = 1/10$
Diffusion steps: $T = 10$
Noise schedule: linear interpolation between 0 and $h$
Number of particles: $N = 100$
Number of interim samples: $M = 4000$ |
| Ours without interaction | Same as *Ours*, except:
    Number of particles: $N = 1$
    Number of independent chains: 100 |

Table 2: Parameter setting for the experiment in Section 5.2.

| Method | Parameters |
|---|---|
| In-and-Out | Initial distribution: $x_0^{(i)} \sim \mathcal{N}(0, I_d)$
Step size: $h = 1$
Number of proposals for rejection sampling: 10000
(Particles are discarded if not accepted)
Number of independent chains: 1000 |
| Ours | Initial distribution: $x_0^{(i)} \sim \mathcal{N}(0, I_d)$
Step size: $h = 1$
Diffusion steps: $T = 10$
Noise schedule: linear interpolation from 0.01 to 1
Number of particles: $N = 1000$
Number of interim samples: $M = 300$ |

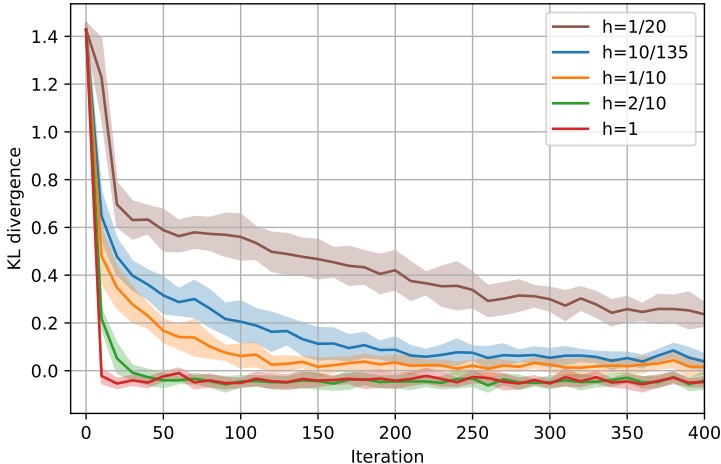

Figure 5: Convergence of KL divergence for different step sizes $h$, with all other parameters set as *Ours* in Table 1. Each curve shows the mean over 10 random seeds, with shaded areas indicating variances. Larger step sizes lead to faster convergence.

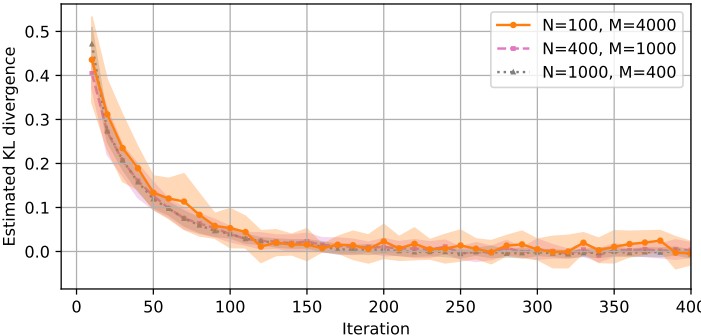

Figure 6: Comparison of hyperparameter tuning while keeping the number of particles used during the algorithm's execution (i.e., $M \times N$). For readability, the first iteration has been omitted. While increasing the number of particles N used for approximating the target distribution stabilizes the KL divergence, changing both M and N results in very similar convergence patterns.

**Effect of step size.** In addition to the default choice $h = 1/10$, we conducted experiments with other values of $h$ (Figure 5). We observe that larger step sizes accelerate convergence, which can be attributed to the heat flow more rapidly bridging the two modes. This phenomenon resembles diffusion-based Monte Carlo methods, where pushforward dynamics from a Gaussian initialization cover the target distribution.

**Hyperparameter sensitivity under fixed computational cost.** Furthermore, Figure 6 shows that when the total number of particles used during the algorithm's execution (i.e., $M \times N$) is fixed, the algorithm exhibits comparable behavior across different choices of $N$ and $M$. Increasing $N$ improves the approximation of the target distribution, thereby stabilizing the estimated KL divergence. Conversely, increasing $M$ enhances the Monte Carlo estimation of the diffusion scores. Thus, under a fixed computational budget, the method remains robust to these hyperparameters.

