# OpenReview forum: "Alternating Diffusion for Proximal Sampling with Zeroth Order Queries"
_ICLR.cc/2026/Conference — ICLR 2026 Poster_

### Official Review · Reviewer_RPkJ · 2025-10-28

**Soundness:** 4
**Presentation:** 4
**Contribution:** 4
**Rating:** 8
**Confidence:** 4

**Summary:**

Method for non-parametric diffusion based approximation of proximal sampling exploiting zeroth-order information
with particle interaction. The authors use a Gaussian mixture surrogate for denoising the particles in the alternating proximal sampling process. They avoid the computation of gradient of $f$ for guidance in the SDEs and use of wasteful rejection sampling for denoising step. This leads to a multi particle sampling algorithm with deterministic runtime and exponential convergence guarantees with accurate score function esitmation

**Strengths:**

The authors present a very clear motivation and methodology for their work. The background is covered in a structure manner and the theoretical analysis and the experiments support the claims in the paper. The paper is very well written and the contributions are noteworthy.

The proposed sampling method uses simple surrogate to bypass inefficient use of rejection sampling and approximation of the score function. The method relies on noisy approximation of the score function of the denoising SDE, which derives it from zeroth order information and multi-particle interactions, improving the coverage of the disconnected modes of the distributions, and alleviating the need for projection and convex support.

**Weaknesses:**

The introduction stresses that the proximal algorithm should be scalable. However the experiments are limited to small $d$ scenarios. It would be informative to have additional results showing convergence time vs $d$ and compare the algorithm with the baselines.

Typo: 146 $B_t^\leftarrow$ is backward Brownian motion.

**Questions:**

The Monte Carlo estimation of the score function is increasing the overall computation with $M$ - But in the experiment section I didn't understand how the authors used large $M$ and compare to algorithms that don't have that many parallel threads in a fair setup. Can the authors provide a description of the fairness?

Mode coverage in the target is attributed to both the ability to use larger noise scale $h$ and the inverse weight of $x$ in high concentration regions. But the two seem to be in contention in $\hat q_{k+1/2}(\cdot|X_k)$, i.e close to $X_k$ and with small $h$ the component weights in Eq 10 will be smaller. Doesn't increasing $h$ at the same time make the weights more uniform?

Have the authors studied the convergece time vs $M$ and $N$ when $N\times M$ is fixed?

---

> ### Author Response · Authors · 2025-11-25
>
> We thank the reviewer for the careful reading and the constructive feedback.
>
> > Typo
>
> Thank you for pointing out the typo. We have made the necessary corrections.
>
> > I didn't understand how the authors used large $M$ and compare to algorithms that don't have that many parallel threads in a fair setup. Can the authors provide a description of the fairness?
>
> In Section 5, the experiments and Figure 2 use iteration as the horizontal axis.
> This means we disregard both the cost associated with RGO's rejection sampling process until acceptance and the cost involved in our method's parallel computation of M particles.
> In practice, rejection sampling can improve wall-clock time when parallelized.
> However, we are examining the dynamics of proximal sampling to verify that KL convergence occurs across iterations, rather than focusing on such minor implementation details.
>
> > close to $X_k$ and with small $h$ the component weights in Eq 10 will be smaller. Doesn't increasing $h$ at the same time make the weights more uniform?
>
> The key aspect of our approach involves setting the objective (denormalized) distribution of the reverse diffusion process to a distribution that approximates uniform (by Gaussian mixture $g_N^{k+1/2}$) multiplied by $e^{-f}$, as expressed in Eq 10.
> In proximal sampling, taking a larger step size $h$ leads to greater exploration, while making $g_N^{k+1/2}$ more uniformly distributed ensures more accurate exploration.
> These two aspects are consistent.
>
> > Have the authors studied the convergece time vs $M$ and $N$ when $N \times M$ is fixed?
>
> We added experimental results comparing performance with fixed $N \times M$ while varying $M$ and $N$, appending them to the final section of Appendix B. The results remain robust when a fixed computational cost is incurred during the algorithm's execution.
>
> >  the experiments are limited to small  scenarios. It would be informative to have additional results showing convergence time vs  and compare the algorithm with the baselines.
>
> The degradation induced by the curse of dimensionality is well known and arises in both zeroth-order methods and rejection sampling.
> In theoretical work on proximal sampling, Fan et al. (2023), cited in our paper, improved the dependence on the dimension $d$ by employing approximated RGO.
> However, their empirical behavior remains unclear, and our experimental setup is consistent with prior studies such as Liang and Chen (2023b) and He et al. (2024).
> Understanding the dimensional dependence of our method and exploring potential improvements in high-dimensional settings remain important future directions, and we plan to document these limitations in Section 7.

---

### Official Review · Reviewer_rzzM · 2025-10-31

**Soundness:** 3
**Presentation:** 3
**Contribution:** 3
**Rating:** 6
**Confidence:** 3

**Summary:**

This paper addresses a canonical isoperimetric sampling problem via a localized reverse heat flow and develops theoretical guarantees by leveraging the proximal-sampling analysis paradigm, reflecting the close connection between proximal operators and localized reverse diffusion. Unlike standard proximal samplers, the proposed method does not require first-order (gradient) information of the target distribution. This is enabled by a Tweedie-style identity that reformulates the denoising score, thereby removing the need for explicit gradients. While related identities have been used for importance weighting in [huang2024a], the probability path considered here is distinct—combining a global proximal path with a local reverse diffusion path—and the paper analyzes a Sequential Monte Carlo (SMC) scheme, which was not studied in [huang2024a].

**Strengths:**

1. Although many components (proximal samplers, reverse diffusion Monte Carlo, and SMC) are drawn from prior work, the paper integrates them coherently to deliver a zero-order sampling scheme with provable convergence. The design is guided by a clear insight: Langevin-based methods inherently require gradients, whereas reverse-diffusion samplers can operate using only zero-order information.
2. The paper explicitly elucidates the close connection between proximal sampling and reverse diffusion, an important and conceptually insightful observation that may inform the design of gradient-free samplers.
3. The exposition is clear and well-structured: the intuition, assumptions, and main theorems are presented transparently, and the experiments are comprehensive, effectively demonstrating the proposed method’s efficacy.

**Weaknesses:**

1. The complexity analysis appears incomplete. The paper provides a one-step KL contraction bound with an additive term, but lacks an end-to-end error and cost analysis that composes these bounds over the full trajectory. A global, non-asymptotic complexity guarantee would substantially strengthen the contribution.
2. The theoretical results are largely asymptotic. In practice, performance with finite $M$ (particles) and $N$ (time steps) is crucial. The assumptions used to control the one-step error via $M$ and $N$ seem idealized; we recommend validating their plausibility empirically on synthetic data and reporting sensitivity to $M$ and $N$.
3. The comparison with RGO is unclear. What constitutes an ``iteration''? Does one pass that updates the entire sequence $\{x_i\}_{i=1}^N$ count as a single iteration or as $N$ iterations? Please clarify the accounting, and provide convergence comparisons against wall-clock time to enable fair, hardware-agnostic evaluation.

**Questions:**

Refer to the above parts.

---

> ### Author Response · Authors · 2025-11-25
>
> We appreciate the reviewer’s feedback.
>
> > The complexity analysis appears incomplete.
>
> > The theoretical results are largely asymptotic.
>
> Our method not only converges asymptotically as $K, T, N, M \to \infty$, but also has been evaluated the order of convergence under certain assumptions.
> These are mentioned in the text and the appendix, but we plan to improve the manuscript by presenting them more explicitly.
>
> > we recommend validating their plausibility empirically on synthetic data and reporting sensitivity to $M$ and $N$
>
> This observation also relates to Reviewer RPkJ's question.
> We added experimental results comparing performance with fixed $M \times N$ while varying $M$ and $N$, appending them to the final paragraph of Appendix B. The results remain robust when a fixed computational cost is incurred during the algorithm's execution.
>
> > The comparison with RGO is unclear.
>
> We describe the experimental setup for comparison with RGO in Section 5 and Appendix B.
>
> > What constitutes an ``iteration''? Does one pass that updates the entire sequence $\{x_i\}_{i=1}^N$ count as a single iteration or as $N$ iterations?
>
> We conducted fair comparisons in the following sense.
> Notably, comparing our method with $N=1$ running 100 independent chains against our method with $N=100$ particles essentially tests the effect of particle interactions.
> Furthermore, as noted in l360, our method approximates the backward diffusion process with 10 steps. Therefore, we are comparing 10 updates of RGO against one iteration of our method, as explicitly stated in Figure 4 with the label (thinning=10).

---

### Official Review · Reviewer_zSYe · 2025-11-01

**Soundness:** 4
**Presentation:** 3
**Contribution:** 2
**Rating:** 6
**Confidence:** 4

**Summary:**

The method provides an alternate implementation of the proximal sampler/RGO, based on an estimate for the scores using a particle ensemble of samples. This method yields a score estimator which appears as a mixture of Gaussians; in particular, the score estimator is implementable using zeroth order oracle queries. Theoretical analysis for this algorithm is provided, showing rates polynomial in the problem parameters. Experimental evidence then shows that this algorithm outperforms standard implementations of the RGO several benchmarks of interest.

**Strengths:**

The method provides a means for implementing the RGO in the proximal sampler using only \emph{zeroth} order queries, which is a more general computational model than the standard gradient oracle model. Furthermore, it does not need any convoluted tricks (MALA + underdamped) to implement the proximal sampler, compared to prior theoretical proposals.

The method appears to work extremely well in practice when compared to the standard implementation of the RGO. Generally, I suppose it is not too surprising that these particle methods can perform well in practice, although theoretical guarantees may be harder to establish.

The method seems to be easily parallelizable.

**Weaknesses:**

The theoretical guarantees are not particularly strong; I would be surprised if in the LSI setting this could improve upon the guarantees for the usual implementation of the RGO. Indeed, as the error scales as 1/N in the number of particles, so we should expect $N \asymp \varepsilon^{-2}$ or polynomial in the accuracy (compared to the standard implementation). Of course, there are other drawbacks to the theory.

**Questions:**

For this algorithm, when running the particle ensemble, do we take the entire ensemble of N particles as samples? This has the risk of inducing extra errors from correlation, but is unlikely to matter much in practice.

I am also surprised that we only see $1/N$ in the error, compared to the usual curse of dimensionality for non-parametric estimation. What is the intuition here?

Line 373 has some space formatting issues.


Notation for R\’enyi divergence switches between $\mathcal R$ and $R$.

---

> ### Author Response · Authors · 2025-11-25
>
> We thank the reviewer for the thoughtful comments and constructive suggestions.
>
> > The theoretical guarantees are not particularly strong; I would be surprised if in the LSI setting this could improve
>
> We show that even with tolerance for score/discritization errors in the diffusion process, the outer proximal sampling iterations converge to the target distribution.
> At the same time, our experiments indicate that the method can operate with large step sizes $h$, indicating that the behavior may be dominated by the inner diffusion-based Monte Carlo component, making it effectively closer to diffusion-based Monte Carlo methods.
>
> > we should expect $N \asymp ε^{-2}$ or polynomial in the accuracy (compared to the standard implementation)
>
> In our error analysis, under certain assumptions, we obtain an $ O(1/N + 1/M) $ bound on the score-estimation error at each point.
> The algorithm not only constructs a surrogate distribution that approximates the target distribution using $N$ particles, but also performs Monte Carlo estimation of the scores with respect to a perturbed version of this surrogate distribution.
> This differs from the setting studied in kernel density estimation, where the goal is to approximate the target density itself using $N$ components.
>
> > do we take the entire ensemble of N particles as samples? This has the risk of inducing extra errors from correlation,
>
> As shown in experiments, our multi-particle system with $N$ samples encourages mixing. In theory, the errors $O(1/N)$, meaning that while finite particles may exhibit correlation-based errors, the bias disappears as $N\to\infty$.
>
> >  the usual curse of dimensionality for non-parametric estimation. What is the intuition here?
>
> We derive the error bound as $O(1/N + 1/M)$ under several distribution-related constants, including the fourth moment of the random variable's norm.
> These constants may worsen with dimensionality, but we evaluated the order of convergence with respect to $N$ and $M$ when the distribution is fixed.
>
> > space formatting issues
>
> > Notation for R\’enyi divergence
>
> Thank you for pointing out the typos. We have made the necessary corrections.

---

### Official Review · Reviewer_bjDQ · 2025-11-01

**Soundness:** 3
**Presentation:** 3
**Contribution:** 2
**Rating:** 6
**Confidence:** 5

**Summary:**

This paper studies proximal sampling for a distribution satisfying the log-Sobolev inequality without implementing the restricted Gaussian oracle (RGO) by rejection sampling. Instead, it uses the idea of diffusion-based Monte Carlo to simulate the backward SDE (i.e., eq (5)) for the backward step in proximal sampling, that is, the RGO. By tracking $N$ particles, the proposed method approximates $\pi^X$ by (9), and hence approximates the score function in (5) by (11). The key idea behind the approximation (9) is importance sampling. Thus, different from the exact implementation of RGO using first-order oracle of potential $f$ (for solving an optimization problem) and rejection sampling, Algorithm 1 approximately implements RGO by simulating (5) over $T$ internal steps within the stepsize $h$ of the (outer) proximal sampling.

The paper presents a complete analysis of Algorithm 1 with both discretization error and score estimation/Monte Carlo simulation error. It also provides two numerical experiments: 1) Gaussian-LASSO mixture; and 2) uniform sampling over non-convex sets. Numerical results in Section 5.1 demonstrate the advantages of the proposed method: it allows the use of a larger stepsize $h$ compared with proximal sampling based on rejection sampling, leading to faster convergence.

**Strengths:**

Unlike the proximal sampling based on rejection sampling, this paper approximately implements RGO via simulation backward SDE and score function using a particle system (i.e., Monte Carlo simulation). The advantage (as reflected in the experiments) is that the stepsize $h$ of the proximal sampling could be taken large (as long as $T={\cal O}(h)$). In particular, $h$ does not depend on dimension $d$ and properties of $f$ such as smoothness $L$. Moreover, the method does not require the first-order oracle of $f$, such as a subgradient; instead, it only assumes the zeroth-order oracle.

**Weaknesses:**

A major shortcoming is that the technical challenge addressed by the work is not very clear. Simulating SDEs with particle systems is a well-studied idea that has appeared frequently in the literature, so the methodological novelty seems limited, in view of the comparison with related works in Section 6. Moreover, the technical depth of the analysis is uncertain, as many of the proof techniques appear to be adapted from existing works, such as Vempala and Wibisono (2019).

Another notable shortcoming lies in the experiment in Section 5.2. In-and-Out finds T1 but fails to reach T2. It is claimed that the proposed method successfully explores both T1 and T2; however, based on Figure 4, it is difficult to conclude that the proposed method clearly identifies either region. The yellow points appear scattered all over, so there is no specific pattern that can be observed. The uniform sampling results are not convincing.

Finally, the comparison only with RGO in Section 5.1 might be limited.

**Questions:**

1. The key idea behind the approximation (9) is importance sampling. It would be nice if this insight could be made explicit in the paper.

2. Typo: line 710, a "$y$" is missing in the Gaussian.

3. Line 130, Fisher information should be relative Fisher information.

4. Some related papers might be missing:
Mixing Time of the Proximal Sampler in Relative Fisher Information via Strong Data Processing Inequality, Andre Wibisono
Proximal Oracles for Optimization and Sampling, Jiaming Liang and Yongxin Chen
Oracle-based Uniform Sampling from Convex Bodies, Thanh Dang and Jiaming Liang

5. Similar to (Kook, Vempala, and Zhang, 2024), the last paper above by Dang and Liang studies proximal sampling for uniform sampling on convex bodies. It presents two implementations of RGO using projection and separation oracles. It would be interesting to compare with this paper as well in Section 5.2.

---

> ### Author Response · Authors · 2025-11-25
>
> We thank the reviewer for the thoughtful comments and constructive suggestions.
>
> > Simulating SDEs with particle systems is a well-studied idea that has appeared frequently in the literature, so the methodological novelty seems limited
>
> Among methods that simulate SDEs using queries of the potential function $f$ without reference samples, simulating the reverse diffusion process under a strictly zeroth-order oracle model is nontrivial and substantially more restrictive.
> Section 6 clarifies the differences between our approach and existing methods, such as ZOD-MC, which is a single-particle method and further relies on additional information, such as the minimizer of $-f$.
> Our contribution is to construct a proximal sampler under the minimal information regime, demonstrating that the dynamics can still be realized under such a limited oracle.
>
> > the comparison only with RGO in Section 5.1 might be limited
>
> Although several works investigate theoretical aspects of proximal sampling, empirical evaluations remain scarce. That said, we compared the proposed method with RGO in the setting of Liang & Chen (2023b), and observed better approximation to the designed dynamics even under weaker queries. We also note that comparisons with samplers that require gradients or other stronger oracles are outside the scope, since our focus is on developing zeroth-order methods.
>
> > there is no specific pattern that can be observed. The uniform sampling results are not convincing.
>
> The table below reports, at iteration $k$, the number of particles located in $T_{1}$, in $T_{2}$, and outside both regions.
> Because the target distribution provides no gradient and no information on distance from the target sets, outward drift is unavoidable (e.g. ULA reduces to pure Brownian motion when $\nabla f \equiv 0$, in which case no spatial structure can emerge).
> By contrast, our method uses membership oracles and succeeds in exploring a nonconvex, disconnected subset of $\mathbb{R}^{3}$, yielding substantially more particles within $T_{1}$ and $T_{2}$ relative to their volumes.
> Practically, our approach can serve as a warm-up. Dynamically reducing $h$ or switching to In-and-Out after a certain iterations may further improve sampling efficiency.
>
> | k | in T_1 | in T_2 | outside |
> |-----:|-------------:|-------------:|--------:|
> | 3    | 769          | 0            | 231     |
> | 10   | 728          | 4            | 268     |
> | 200  | 635          | 35           | 330     |
> | 400  | 613          | 46           | 341     |
>
> > Q1
>
> We have added a remark at the end of the derivation in Appendix A.1, explaining the rationale behind using this surrogate distribution by importance sampling.
>
> > Q2,3
>
> Thank you for pointing out the typos. We have made the necessary corrections.
>
> > Q4,5
>
> Thank you for suggesting these references. We will cite them in the revision. Their relationship to our approach is summarized below. Regarding the first reference, the assumptions and metrics used in theoretical analysis differ from ours. The second and third references were recently submitted/revised after our submission, and our work does not build upon them.
> As mentioned in Section 5.2, projection from any point outside the target domain constitutes stronger information about the target domain and is feasible only in limited cases. Our approach and In-and-Out operate based solely on membership queries, without requiring such detailed information.

---

> > ### Comment · Reviewer_bjDQ · 2025-11-26
> >
> > The reviewer appreciates the authors' response. The explanation for Figure 4 was not convincing. Moreover, it is hard to tell whether the proposed method in this paper is better than RGO from Figure 3. In the right plot of Figure 3, RGO appears to match the histogram more closely. Given the above reasons, the reviewer decided to maintain the score.

---

> > > ### Author Response · Authors · 2025-11-29
> > > **Additional Notes on Experimental Figures**
> > >
> > > Thank you for your feedback regarding the display of experimental results.
> > >
> > > > it is hard to tell whether the proposed method in this paper is better than RGO from Figure 3. In the right plot of Figure 3, RGO appears to match the histogram more closely
> > >
> > > In Figure 3 (right), the RGO histogram is shown as a reference obtained from a sufficiently long run, whereas our method reaches its results with significantly fewer iterations. This means we paid much higher compute for the RGO reference.
> > > Since the Gaussian-Lasso mixture cannot be sampled directly, Liang and Chen (2023b) provided only qualitative evidence that an RGO-based proximal sampler, when run for 500,000 iterations, yields a one-dimensional marginal that closely matches this reference.
> > > To estimate the KL divergence and quantify convergence, we require reference particles that approximate the continuous target accurately.
> > > We therefore use the long-run output of the RGO-based sampler as the reference and confirm its adequacy in the right panel of Figure 3.
> > > Although this construction is described in Section 5.1 and Appendix B, we will make it explicit in the caption of Figure 3.
> > >
> > > > explanation for Figure 4 was not convincing
> > >
> > > In Figure 4, particles inside the two toroidal regions in $\mathbb{R}^{3}$ are shown in dark purple, but points that lie inside the torus along the depth direction (i.e., in front of or behind the surface) are projected as outside, which increases the number of apparently outer points in the two-dimensional plot.
> > > Particles slightly outside the surfaces or between the two tori appear because the reverse diffusion process targets an approximate surrogate distribution rather than enforcing a hard threshold as in rejection sampling.
> > > This leads to the exploratory behaviour induced by the importance-weighted surrogate defined in Eq. 10, and is related to your Q1.
> > > The experiment is a purely synthetic setup intended to illustrate this effect and is not meant to imply that our method alone is optimal for this particular target.

---

### Meta-Review · Area_Chair_meRw · 2026-01-09

**Summary:**

The submission proposes an alternating diffusion view of proximal sampling that yields a learning-free, zeroth-order approximate proximal sampler with fixed runtime. The key technical idea is to approximate the intermediate distribution with a particle-based Gaussian-mixture surrogate, enabling Monte Carlo score estimates to simulate the reverse-time dynamics without requiring gradients or costly rejection sampling, and allowing for larger step sizes. The paper provides supporting theory by extending proximal-sampling analyses to show that, assuming log Sobolev inequality, the method retains contraction properties up to discretization and score-estimation errors. Empirically, results on a Gaussian–LASSO mixture indicate faster convergence in iteration count compared with an RGO-based proximal baseline, and ablations suggest multi-particle interaction is important; a nonconvex/disconnected-set uniform-sampling example further illustrates the method’s potential in challenging settings.

Reviewers are generally positive about the conceptual integration and the practical value of a strictly zeroth-order proximal-style sampler, though some concerns remain about technical novelty, evaluation clarity and fairness, and theoretical analyses. Overall, I recommend to accept the paper.

**Reviewer Concerns:**

The reviewers are concerned about (i) novelty, as related particle-SDE ideas exist and the contribution is best viewed as a meaningful instantiation in a more restrictive oracle model; (ii) evaluation clarity and fairness, since comparisons are primarily iteration-based and do not fully normalize for Monte Carlo/parallel compute budgets; and (iii) theory presentation, where an end-to-end non-asymptotic complexity bound would strengthen the claims. Overall, the reviewers feel that the contribution is already good enough for an ICLR paper.

The rebuttal clarified the oracle model distinctions, iteration accounting, and added additional analyses/ablations (including sensitivity to particle/MC budgets), which addresses some minor methodological and presentation concerns.

**Reviewer Scores:**

The reviewers' scores would not increase, after the rebuttal.

---

### Decision · Program_Chairs · 2026-01-26

Accept (Poster)